# Efficient and sustainable water electrolysis achieved by excess electron reservoir enabling charge replenishment to catalysts

Gyu Rac Lee[1,7], Jun Kim[2,7], Doosun Hong[3,7], Ye Ji Kim[1,4], Hanhwi Jang [1], Hyeuk Jin Han[5], Chang-Kyu Hwang[6], Donghun Kim [3] ✉, Jin Young Kim [2] ✉ & Yeon Sik Jung [1] ✉

Suppressing the oxidation of active-Ir(III) in $IrO_x$ catalysts is highly desirable to realize an efficient and durable oxygen evolution reaction in water electrolysis. Although charge replenishment from supports can be effective in preventing the oxidation of $IrO_x$ catalysts, most supports have inherently limited charge transfer capability. Here, we demonstrate that an excess electron reservoir, which is a charged oxygen species, incorporated in antimony-doped tin oxide supports can effectively control the Ir oxidation states by boosting the charge donations to $IrO_x$ catalysts. Both computational and experimental analyses reveal that the promoted charge transfer driven by excess electron reservoir is the key parameter for stabilizing the active-Ir(III) in $IrO_x$ catalysts. When used in a polymer electrolyte membrane water electrolyzer, Ir catalyst on excess electron reservoir incorporated support exhibited 75 times higher mass activity than commercial nanoparticle-based catalysts and outstanding long-term stability for 250 h with a marginal degradation under a water-splitting current of $1 \, A \, cm^{-2}$. Moreover, Ir-specific power ($74.8 \, kW \, g^{-1}$) indicates its remarkable potential for realizing gigawatt-scale $H_2$ production for the first time.

Since hydrogen energy has emerged as a viable alternative under the global environmental crisis due to its zero-emission nature[1,2], researchers have devoted extensive efforts over the past several decades toward realizing carbon-free production of hydrogen[3]. More recently, economical and scalable hydrogen production using surplus electricity generated from renewable energy sources has been attempted[4–6]. In particular, polymer electrolyte membrane water electrolysis (PEMWE) has been explored as a promising candidate for this due to its superior energy efficiency, purity of the

generated hydrogen, and ability to operate at high current densities[7]. However, the commercialization of PEMWE is being hampered by the large overpotential and inevitable use of a substantial amount of high-cost noble metals, which are deeply associated with the sluggish kinetics of the oxygen evolution reaction (OER) in acidic conditions[8]. Furthermore, in order to meet the high standards of industrial applications, a large loading amount (usually over at least $0.5 \, mg \, cm^{-2}$) of Ir is required, greatly increasing the total cost[4,9]. To address these issues, extensive studies have been

[1]Department of Materials Science and Engineering, Korea Advanced Institute of Science and Technology, 291 Daehak-ro, Yuseong-gu Daejeon 34141, Republic of Korea. [2]Hydrogen·Fuel Cell Research Center, Korea Institute of Science and Technology, 14-gil 5, Hwarang-ro, Seongbuk-gu Seoul 02792, Republic of Korea. [3]Computational Science Research Center, Korea Institute of Science and Technology, 14-gil 5, Hwarang-ro, Seongbuk-gu Seoul 02792, Republic of Korea. [4]Department of Materials Science and Engineering, Massachusetts Institute of Technology, Cambridge, MA 02139, USA. [5]Department of Environment and Energy Engineering, Sungshin Women's University, 55, Dobong-ro 76ga-gil, Gangbuk-gu Seoul 01133, Republic of Korea. [6]Materials Architecturing Research Center, Korea Institute of Science and Technology (KIST), 14-gil 5, Hwarang-ro, Seongbuk-gu Seoul 02792, Republic of Korea. [7]These authors contributed equally: Gyu Rac Lee, Jun Kim, Doosun Hong. ✉e-mail: donghun@kist.re.kr; jinykim@kist.re.kr; ysjung@kaist.ac.kr

conducted to enhance both the activity and stability of Ir-based catalysts while minimizing the use of Ir[10–14].

IrO$_x$, which is typically formed during initial activation cycling of Ir catalysts, is known to exhibit higher activity toward the OER compared to metallic Ir or IrO$_2$ due to its multiple valence states[15–17]. In particular, a linear correlation was observed between the catalytic activity of IrO$_x$ and the fraction of Ir(III) species[15,18]. However, due to the electrochemical nature of the OER, Ir(III) tends to be oxidized to Ir(IV) or Ir(V) as the reaction proceeds. During the oxidation of Ir(III), the formation of soluble intermediates leads to gradual Ir dissolution and degradation of the catalytic performance over time[15,19,20]. The previous approaches to develop Ir-based electrocatalysts have mainly been focused on enhancing activity, without sufficiently reflecting catalyst deterioration and the durability issue[21]. Alternatively, sustainably retaining active-Ir(III) species is considered one of the most desired strategies to overcome the fundamental trade-off relationship between activity and durability[22,23], but remains a difficult challenge to realize under the harsh electrochemical conditions of OER.

With these motivations, efforts were recently directed to identifying suitable metal oxide supports that could offer stronger interactions with catalysts and, as a result, enhanced charge transfer[17,24–26]. Charge transfer from the metal oxide supports to Ir catalysts can be associated with improvements in both OER activity and durability because a higher fraction of Ir(III) species would be maintained by the constant replenishment of charge from the supports during the reaction[27]. Indeed, attempts to control metal-support interactions to enhance OER performance were reported in only a few studies, and they were based on complicated compositions or complex metal oxides composites such as TiON, Nb$_{0.05}$Ti$_{0.95}$O$_2$, and TiO$_2$-MoO$_x$[28–30]. This is because common metal oxides have difficulty in overcoming the limited charge transfer capability due to their innate electronic structure[31,32]. Moreover, unfortunately, enhanced charge transfer capability of these convoluted materials is observed only for very restricted complex compositions. Recently, an additional defect engineering approach for metal oxide supports was also reported, where the formation of oxygen vacancies (V$_O$) was effective in reinforcing the charge transfer capability by increasing the number of electrons in the entire system[33]. However, cation leaching, which occurs mainly due to the slow compensation rate of V$_O$, leads to structural collapse[34–36]. Overall, these previous studies suggest the need to develop a more universal catalyst support design that can more sustainably boost the charge transfer from metal oxide supports to IrO$_x$ and, as a result, stabilize Ir(III) species without structural deformation.

In this study, we introduce a unique support modification strategy, namely, an excess electron reservoir (EER), which serves as an electron-donating layer, formed on a metal oxide support to preserve Ir(III) states via charge replenishment. Using a model system composed of an IrO$_x$ catalyst and antimony-doped tin oxide (ATO) as a base support material, we identified charged-oxygen groups (O$^-$ and O$_2^-$) as an optimal EER material. They are thermodynamically stable and could potentially maximize charge donation to IrO$_x$. O$_2^-$ layers were experimentally in situ formed on ATO support under oxygen-rich evaporative deposition conditions of ATO. The Ir catalysts placed on EER-incorporated ATO demonstrated significantly enhanced mass activity, around 75 times higher than those of commercial Ir nanoparticle catalysts, and long-term stability over 250 h at 1.0 A cm$^{-2}$ current density (0.624 mV/h) in a PEMWE single cell. Our results clearly overcome the hitherto unresolved trade-off relationships between catalytic activity and durability in PEMWE. Employing X-ray photoelectron spectroscopy (XPS) and density-functional-theory (DFT) analyses, we reveal that these improvements are associated with stabilization of Ir(III) species due to the promoted charge transfer by the EER.

## Results

### Design and fabrication of EER-contained support material

Figure 1a illustrates the proposed role of the EER by comparing cases with and without EER, which is located between the support and IrO$_x$ catalyst. The structure was designed to transfer more electrons to IrO$_x$ and boost the catalytic performance[15,18]. With the EER residing on top of the support, the electronic interaction between the catalyst and support can be reinforced to allow a high fraction of Ir(III) species in the IrO$_x$ catalyst. First, in order to identify promising candidate materials for the EER, DFT calculation-based screening was performed (Supplementary Fig. 1). We hypothesized that key requirements of the EER are structural stability and enhanced electron donation from the support to IrO$_x$. In this regard, we used three criteria for the DFT screening processes, as depicted in Fig. 1b: EER processability, adsorption energy between EER and underlying ATO (E$_{ads}$), and charge transfer capability between the support and IrO$_x$. In the EER processability stage, we assumed that desirable candidates for the EER should reserve more electrons within, and preferably the constituent elements should be non-metals with high electron affinities. Therefore, nine nonmetallic atomic or diatomic species (B, N, O, F, P, S, N$_2$, O$_2$, and F$_2$) and their appropriate oxidation charge states (between 0 and -3) were adopted for the subsequent DFT modeling. In the next stage, the structural stability of the EER was evaluated. Adsorption energy (E$_{ads}$) calculations (Supplementary Fig. 2) revealed that several extrinsic species including B, N, P, and S are thermodynamically unstable (E$_{ads}$ >0), and as a result not suitable for the EER (Fig. 1c). Lastly, the EER-driven charge transfer ability (Supplementary Figs. 3, 4) was also evaluated, and those with lower transfer values (<0.10 e$^-$) were excluded (Fig. 1c). As a result, charged oxygen species (O$^-$ and O$_2^-$) and F$^-$ finally remained as promising EER candidates. In this work, charged oxygen species are selected because, in order to generate F$^-$ on the surface of ATO, additional processes such as reactive-ion plasma treatment must be accompanied. In contrast, charged-oxygen groups can be formed in situ during the process of ATO deposition[37], and the detailed fabrication process will be described below.

The XPS depth profile of an e-beam deposited SnO$_x$ thin film with ~50 nm thickness on the Si substrate exhibited that the atomic ratio of O and Sn changes along the depth of the thin film (Fig. 1d). In particular, there is an oxygen-rich bottom surface with a thickness of about 5 nm, which indicates that the initially deposited SnO$_x$ thin film has a high O/Sn ratio. E-beam irradiation can reduce metal oxides[37,38], which causes dissociation of the SnO$_2$ source into two elemental species (O and Sn) with distinct volatilities and kinetics. Thus, high oxygen composition is attained at the initial stage of the film deposition until equilibrium is reached. Different from bulk SnO$_2$, e-beam evaporation-deposited SnO$_x$ demonstrated a Raman peak emerging at around 1100–1200 cm$^{-1}$, corresponding to O$_2^-$ (Fig. 1e)[39,40], which is one of the candidates for EER deduced from the DFT calculations. Moreover, the atomic ratio of O and Sn in deposited SnO$_x$ can be controlled by selecting an initiation point of e-beam deposition, as shown in Fig. 1d, enabling the fabrication of SnO$_x$ with varied EER content. Based on this EER-incorporated SnO$_x$, Sb was additionally added as an n-dopant to enhance electrical conductivity. XPS analysis was performed to quantify the oxygen contents and analyze their chemical states of ATO with three different relative densities of EER (Supplementary Fig. 5): sparse, moderate, and dense EER ATO (S-ATO, M-ATO, and D-ATO). The O/Sn and Sn$^{Satellite}$/Sn$^{IV}$ ratios can experimentally distinguish the S-ATO, M-ATO, and D-ATO samples. It can be seen that as more EER is incorporated, the intensity of the oxygen peak becomes higher compared to the Sn peak and the ratio of O$_2^-$ to lattice oxygen is also higher, in comparison with the bulk sample. Indeed, D-ATO presents the highest O/Sn ratio of 0.94, which is larger than the values of 0.569 for M-ATO and 0.401 for S-ATO. In addition, it can be seen that as more EER is contained, the ratio of Sn$^{Satellite}$/Sn$^{IV}$ obtained through deconvolution of the Sn 3$d$ spectra is higher (Supplementary Fig. 6). These results

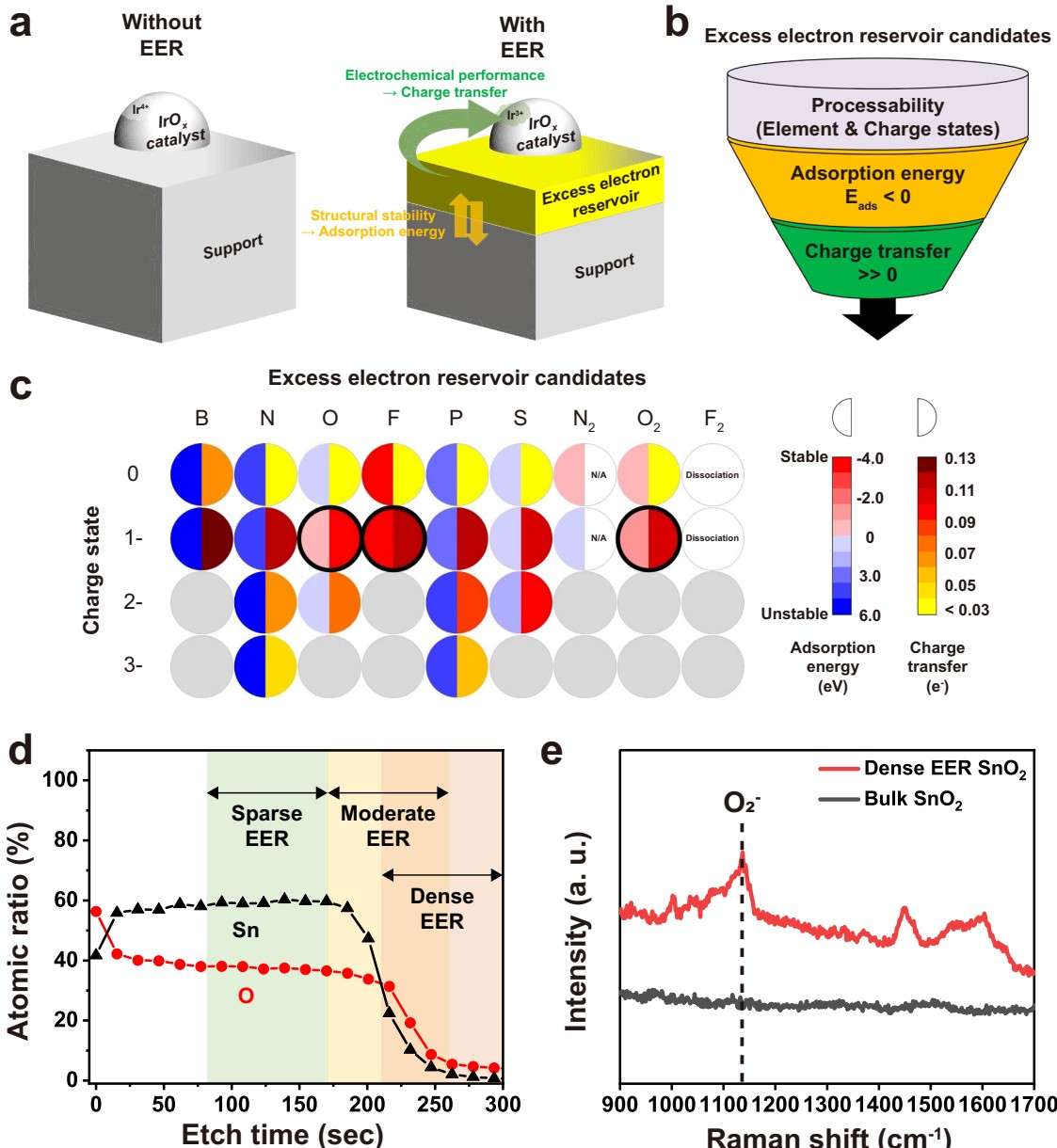

**Fig. 1 | Design and prediction of the excess electron reservoir (EER)-contained support that enables Ir$^{3+}$ stabilization based on computational screening and experimental fabrication strategy of EER using superoxide (O$_2^-$). a** Schematic illustration of the catalyst-support interaction with and without EER. **b** DFT screening process to search appropriate EER candidates, which is composed of three steps (processability, adsorption energy, and charge transfer). **c** Adsorption energy and charge transfer values predicted by the screening process. The term dissociation in the F$_2$ column refers to cases where the adsorbed F$_2$ molecule is dissociated into F atomic species upon DFT relaxations. The term N/A in the N$_2$ column refers to cases where N$_2$ molecule is adsorbed too far (>4 Å) from the support surface to serve as an EER. **d** XPS depth profile of an e-beam deposited SnO$_x$ thin film on Si substrate. The three regions are divided based on the atomic percentage of O and Sn: red (where the atomic percentage of O is higher than Sn), yellow (where the atomic percentage of O intersects with Sn), and green (where the atomic percentage values of O and Sn are comparable). **e** Raman spectroscopy spectra obtained from dense EER SnO$_2$ and bulk SnO$_2$. The peak located at around 1150 cm$^{-1}$ indicates a superoxide. Note that a. u. represents arbitrary units.

indicate that the O$_2^-$-based EER can be obtained with one-step deposition of SnO$_x$ without the need of an additional treatment or deposition process.

### Fabrication and characterization of Ir/D-ATO

We now demonstrate ATO with dense EER at the surface as a support material for high performance Ir-based electrocatalysts (Ir/D-ATO). In Fig. 2a, the catalyst system consisting of Ir, ATO, and dense EER at the surface is illustrated. To achieve a high surface area to volume ratio and efficient mass transport, 3D-nanostructured Ir/D-ATO was prepared by a solvent-assisted nanotransfer printing method (S-nTP), as

we reported previously (Fig. 2b)[41]. Sequential angle deposition and high-temperature annealing at ambient conditions were used during S-nTP to fabricate the basic building blocks of Ir/D-ATO. All the components (Ir, SnO$_2$, Sb) constituting the Ir/D-ATO, including the dense EER at the surface of ATO, are formed sequentially through the oblique-angle deposition process. After the transfer printing process, the Ir/D-ATO was annealed at 700 °C in an Ar atmosphere for several purposes: Sb-doping into the SnO$_2$ matrix, stabilization of Ir at the support, and crystallization. In addition, during the high-temperature annealing process, possible organic residues on the Ir/D-ATO can be removed. First, an XPS analysis was conducted to identify the bonding

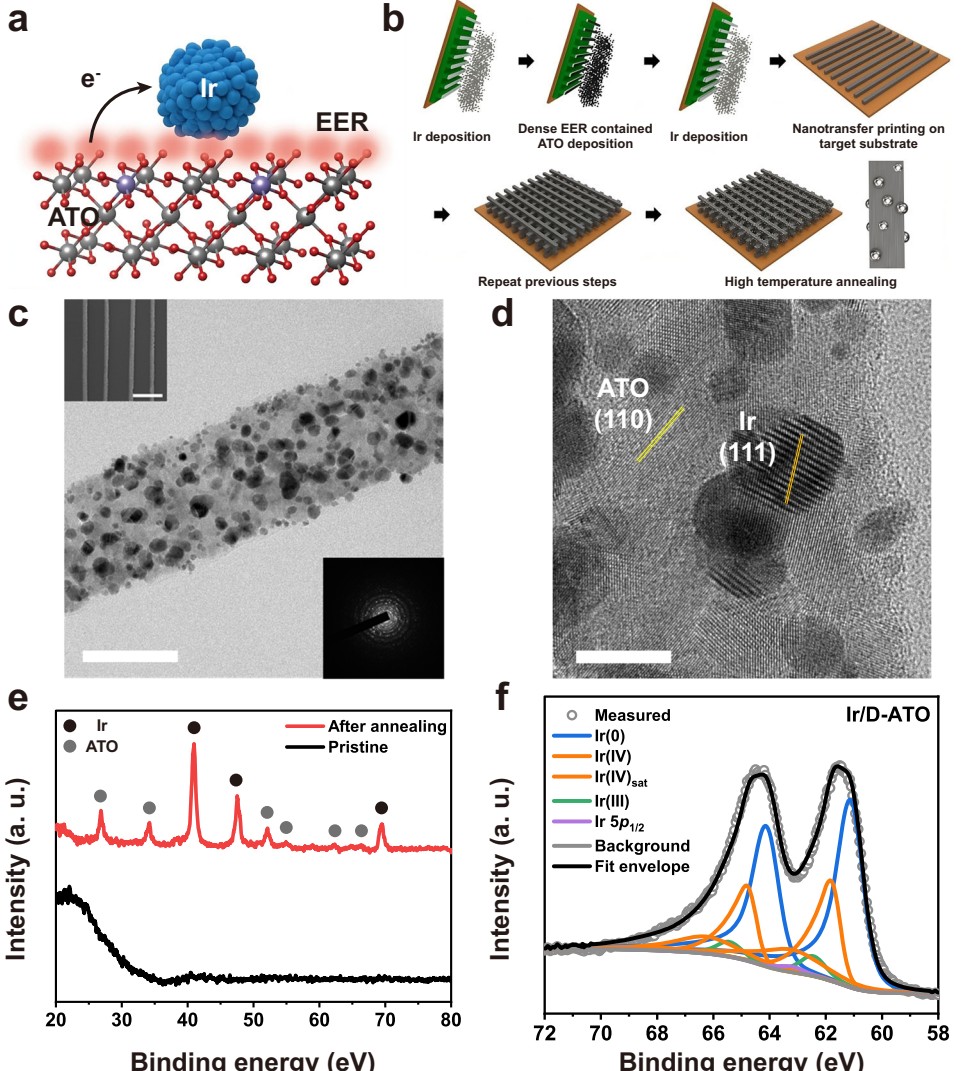

**Fig. 2 | Fabrication and characterization of dense EER ATO-supported Ir catalyst (Ir/D-ATO). a** Schematic illustration of Ir/D-ATO for OER. **b** Fabrication process of Ir/D-ATO using solvent-assisted nanotransfer printing (S-nTP) with sequential e-beam deposition and high-temperature annealing. **c** TEM image with 50 nm scale bar (the inset images: SEM image with 500 nm scale bar (upper left) and SAED pattern (lower right) of Ir-/D-ATO) and **d** HRTEM image with 5 nm scale bar of the fabricated Ir/D-ATO. **e** XRD spectra of Ir/D-ATO before and after high-temperature annealing. **f** XPS spectra of the Ir 4*f* level on Ir/D-ATO. Note that a. u. represents arbitrary units.

states of Sb in the $SnO_2$ matrix. Successful doping of Sb in the $SnO_2$ matrix was confirmed by the shift of Sb 3*d* peaks to higher binding energy (-542 eV ($3d_{5/2}$)) after annealing (Supplementary Fig. 7)[42].

The transmission electron microscopy (TEM) image shows that Ir nanoparticles with a diameter of 2–10 nm are well distributed on the surface of the D-ATO support (Fig. 2c). The energy-dispersive X-ray spectroscopy (EDS) mapping images also exhibit that all the elements such as Sn, Sb, and Ir are uniformly present throughout the whole nanowires (Supplementary Fig. 8). As shown in the selected area electron diffraction (SAED) pattern and the high-resolution TEM (HRTEM) image (right lower inset image in Fig. 2c, d), the crystallinity of the Ir catalysts and D-ATO support improved after thermal annealing. In particular, most of the exposed crystal planes are mainly composed of Ir (111) and ATO (110) planes. These results are consistent with the X-ray diffraction (XRD) measurement, as shown in Fig. 2e. In the XRD spectra, the sharp and narrow peaks at the thermally annealed Ir/D-ATO exhibited enhanced crystallinity compared to the pristine sample. The peak positions of Ir and ATO in Ir/D-ATO are coincident with those of references such as bulk ATO, Ir/C, and Ir black powder, respectively (Supplementary Fig. 9). A XPS analysis was carried out to

identify the chemical state of the Ir/D-ATO (Fig. 2f). By deconvoluting the Ir 4*f* peaks into metallic Ir, Ir(III), and Ir(IV), while considering factors such as the Functional Lorentzian (LF) lineshape and Ir 5*p* ½ peak, it was confirmed that the fractions of each component were 55.10%, 4.92%, and 39.28%, respectively[43,44]. Most of the Ir nanoparticles in the as-prepared Ir/D-ATO existed in the form of metallic Ir or Ir(IV), and the fraction of Ir(III) was significantly lower.

The final morphology of the basic building block−Ir/D-ATO (width: 50 nm/pitch: 400 nm)−was confirmed through scanning electron microscopy (SEM), as shown in the upper left inset of Fig. 2c. By successively printing the layers of aligned nanowire building blocks with a perpendicular orientation to the previous layer, 3D nanostructures composed of multilayer-stacked Ir/D-ATO nanowires were formed (Supplementary Fig. 10).

## Half-cell OER characterization
To evaluate the electrochemical performance of Ir/D-ATO toward the OER, half-cell measurements were conducted in an Ar-saturated 0.05 M $H_2SO_4$ electrolyte using a rotating disk electrode (RDE) setup. We also measured the OER activity of the commercial catalysts (Ir/C, Ir

black) and Ir catalysts supported on a graphite carbon nanostructure (Ir/CNW). The overall morphology of Ir/CNW is similar to the Ir/D-ATO, except that the material constituting the support is graphite carbon (Supplementary Fig. 11). We measured the electrochemically active surface area (ECSA) of Ir/D-ATO, Ir/CNW, Ir/C, and Ir black through a CO stripping method[45] (Supplementary Fig. 12). As shown in Fig. 3c, Ir/D-ATO exhibited the highest ECSA of ≈132 m² g⁻¹, which is almost 2.7 times higher than that of Ir/C or Ir black (≈49 m² g⁻¹). In addition, Ir/ CNW also showed an enhanced ECSA compared to the nanoparticle-type commercial catalysts. This is because the small amount of Ir

(Supplementary Table 1) is evenly dispersed on the entire 3D-nanostructured supports, which have a fully accessible large surface area due to the entirely connected channel structures.

To compare the OER activities of each catalyst precisely, we measured the current density through linear sweep voltammetry (LSV) (Fig. 3a) and calculated the mass activity by normalizing the current density values at 1.55 $V_{RHE}$ with the Ir loading amount and the specific activity by normalizing the mass activity with the ECSA of the catalysts, respectively. The amount of Ir loading in Ir/D-ATO was determined through inductively coupled plasma mass spectrometry (ICP-MS)

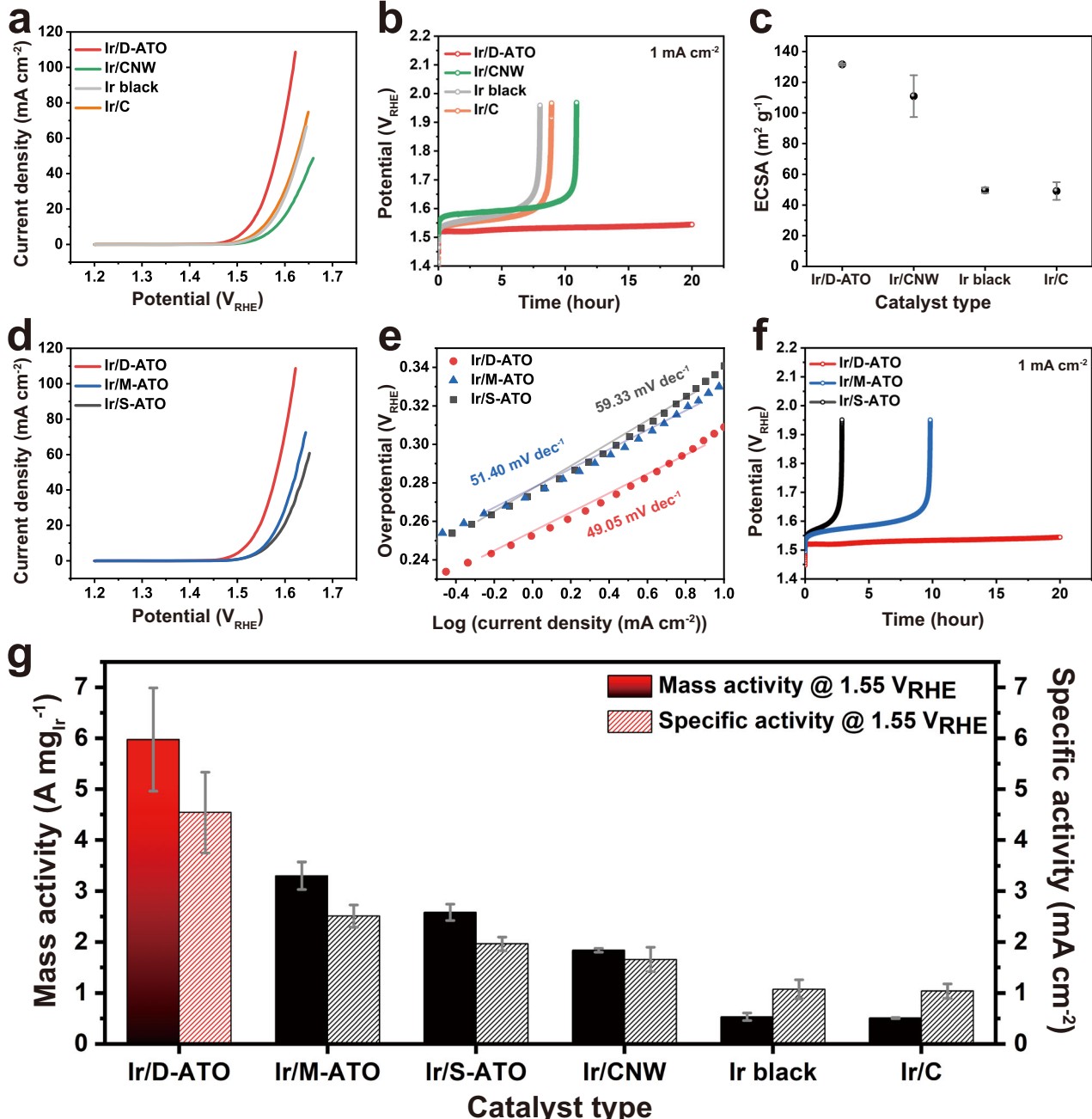

**Fig. 3 | Characterization of OER catalytic performance of Ir/D-ATO and other references in half-cell measurement. a** OER polarization curves and **b** Chronopotentiometry measurements data of the Ir/D-ATO (2.023 µgIr cm⁻²) and other references (17.8 µgIr cm⁻²) are compared. **c** The ECSA calculated by CO stripping (Ir/D-ATO, Ir/CNW, Ir black, and Ir/C). The error bars represent the standard deviation of the mean (*n* = 3). Comparison of electrochemical performance of

Ir/D-ATO, Ir/M-ATO, and Ir/S-ATO. **d** OER polarization curves, **e** Tafel plots showing analogous Tafel slopes and **f** chronopotentiometry measurements data. **g** Mass activity and specific activity comparison of Ir/D-ATO and others. All values represent the mean ± standard deviation (*n* = 3). A 0.05 M $H_2SO_4$ electrolyte with 34−37 Ω of electrolyte resistance was used.

using samples formed on Cu foil. The Ir loading amount on Cu foil can accurately represent the amount of Ir loading on the glassy carbon electrode or Nafion membrane. This is due to the absence of damage during the etching of Cu foil and the subsequent transfer processes (Supplementary Fig. 13). The mass activity (5.975 A mg$_{Ir}^{-1}$) of the Ir/D-ATO corresponds to a value almost 11 times higher than the performance (0.533 A mg$_{Ir}^{-1}$) of Ir black and (0.510 A mg$_{Ir}^{-1}$) of Ir/C (Fig. 3g). These mass activity values of the commercial catalysts are comparable to values reported in recent studies[16]. Furthermore, 5.975 A mg$_{Ir}^{-1}$ is the highest mass activity value ever reported for Ir-based catalysts using metal oxide supports (Supplementary Table 2).

However, the large ECSA of Ir/D-ATO alone cannot explain the significant improvement of the mass activity. Therefore, the specific activity representing the intrinsic activity of catalysts should be considered. As shown in Fig. 3g and Supplementary Table 2, Ir/D-ATO also presents superior specific activity compared to the other catalysts and Ir-based catalysts using metal oxide supports reported previously. In particular, the comparison with Ir/CNW using the same 3D nanostructure indicates that there is an additional factor leading to the enhancement of specific activity in addition to the promoted mass transport in our 3D woodpile electrocatalysts[16,46]. In general, it is well known that the interaction between the carbon support and catalysts is too weak to augment the intrinsic activity of catalysts[17]. The specific activity of the Ir/D-ATO was measured to be almost 273% larger than those of Ir/CNW, despite similar ECSA values. These features support that the presence of the EER on the support plays an important role in the superior OER activity of Ir/D-ATO. Moreover, a consistent trend was confirmed in the electrochemical impedance spectroscopy (EIS) analysis at 1.55 V$_{RHE}$ (Supplementary Fig. 14). Both Ir/D-ATO and Ir/CNW exhibited similar Ohmic resistance to that of Ir/C or Ir black, indicating comparable conductivity of the electrons and protons. However, the charge transfer resistance of the Ir/D-ATO is significantly improved compared to other catalysts. This also suggests that the EER on the support can accelerate the overall charge transfer of Ir/D-ATO during the OER process.

We also assessed the long-term stability of the Ir/D-ATO by measuring the potential to obtain constant current density together with the reference samples. As shown in Fig. 3b, degradation of the catalytic activity is indicated by a gradual increase in potential. During the stability test, the potential for carbon-supported Ir catalysts (Ir/C, Ir/CNW) sharply increased after 8 to 10 h, and for Ir black, the catalytic activity seriously degraded after 7 h. On the contrary, the estimated potential of Ir/D-ATO was maintained over 20 h, which means no degradation occurred during the test. Higher current density (10 mA cm$^{-2}$) was also applied to evaluate the durability of the catalysts (Supplementary Fig. 15). It is well known that the glassy carbon working electrode can also suffer damage at high current density, which affects the stability of catalysts[47,48]. To mitigate carbon corrosion and detachment of Ir catalysts from the backing electrode under harsh conditions, a Ti plate was used as a substrate. When such a high current density was applied to the catalysts, Ir/C degraded almost immediately, and Ir black also lost most of its catalytic activity in less than 3 h. However, Ir/D-ATO showed only a slight increase of potential and maintained its activity for about 18 h. Mostly, degradation of catalysts is caused by dissolution and detachment during the oxidation of the Ir surface[15,49,50]. To quantify the absolute Ir dissolution after the stability measurement, we measured the potential for 15 h at 1 mA cm$^{-2}$. The dissolved Ir in the electrolyte was evaluated using ICP-MS, which confirmed a low Ir mass loss of about 21.3% of the initial Ir loading. Charge transfer of the EER-contained support can prevent the oxidation of Ir catalysts and serves to maintain them, which can effectively suppress the dissolution of Ir, leading to superior durability of Ir/D-ATO. Overall, the enhanced OER performance of Ir/D-ATO can be attributed to the intensified metal-support interaction by the EER.

To elucidate the effect of the EER on the OER performance, three different catalysts (Ir/ATOs) with varying density of EER (dense, moderate, and sparse EER) on the surface of ATO were fabricated: Ir/D-ATO, Ir/M-ATO, and Ir/S-ATO. We confirmed that these three catalysts formed on the different supports have no difference in physical and chemical characteristics. There was no significant dissimilarity in structural morphology, particle size distribution of Ir, and crystallinity, as confirmed through TEM and HRTEM images (Supplementary Fig. 16). It was also confirmed through a XPS analysis (Supplementary Fig. 18) that the chemical state of the as-synthesized catalysts was similar. This means that only the specific activity, not the ECSA, affects the catalytic performance. According to the LSV curves in Fig. 3d, there is a positive correlation between the density of EER contained in the ATO support and the catalytic activity. Also, the Tafel slope of Ir/D-ATO (49.05 mV dec$^{-1}$, Fig. 3e) showed a lower value compared to Ir/M-ATO (51.40 mV dec$^{-1}$) and Ir/S-ATO (59.33 mV dec$^{-1}$), indicating that Ir/D-ATO is more suitable for boosting OER kinetics. The stability of the catalysts was also evaluated based on the content of the EER, and it is found that the trend for the stability is identical to that for the activity (Fig. 3f). These improvements in both the activity and stability with a higher density of EER in ATO strongly indicate a sustainable charge transfer effect between the EER-incorporated support and Ir. A more detailed fundamental analysis to elucidate the relationship between the EER and the OER performance is discussed in the next section.

## Characterization of O$_2^-$ anion-mediated electron transfer

In order to further verify whether the enhanced electrocatalytic activity and durability benefits from the intensified interactions caused by EER, a further XPS analysis was conducted. The Ir 4$f$ spectra were deconvoluted to compare the ratio of metallic Ir, Ir(III), and Ir(IV) of the catalysts attained in each sample: as-synthesized, post-activation, and post-accelerated degradation test (ADT) with Ir/ATOs as well as the other catalysts (Supplementary Figs. 17, 18, and 19)[43]. First, from the comparison of the oxidation states of Ir/D-ATO and Ir/CNW summarized in Supplementary Fig. 20, it was confirmed that, despite that the same method is employed for forming the Ir catalysts, the catalyst-support interaction (CSI) effect significantly depends on the type of support materials. More electrons are supplied from the EER-contained ATO to the catalysts compared to graphite carbon, which leads to a high Ir(III) ratio across all stages of the samples. Moreover, as depicted in Fig. 4a, there is a small variation in the proportion of Ir(III) in the as-synthesized samples, and a relatively higher Ir(III) portion is exhibited for Ir/D-ATO. Furthermore, after the electrochemical activation, the ratio of Ir(III) species increased with the EER content, and Ir/D-ATO exhibited the highest Ir(III) portion (20.69%). In particular, the Ir(III) to Ir(IV) ratio (R$_{III/IV}$) of the Ir/D-ATO was calculated to be 0.563, which corresponds to approximately 1.44 times and 1.77 times that of Ir/M-ATO and Ir/S-ATO, respectively. These results indicate that more electrons are supplied from the supports to the catalysts due to the intensified metal-support interaction induced by the EER, and the amount of charge transfer is proportional to the density of EER. The effects of the EER on the oxidation states of IrO$_x$ can be affirmed more clearly after the ADT. For the case of Ir/D-ATO, not only did the overall portion of Ir(III) increase, but the ratio of Ir(III) to Ir(IV) also significantly rose to 0.757, which is 1.34 times higher than the value of the post-activation sample. On the other hand, the other catalysts with lower EER density exhibited a substantial decrease of Ir(III)/Ir(IV) after the ADT. Finally, Sn 3$d$ and Sb 3$d$ spectra were also analyzed to investigate any changes in the chemical composition of the support after the ADT, and it was verified that there was no noticeable difference, except for a slight shift (0.65 eV) of the Sn 3$d$ peak compared to the initial status (Supplementary Fig. 21).

Besides the XPS analysis, we additionally performed DFT calculations to understand the origin of electron transfer with varying O$_2^-$ concentrations in the EER (Fig. 4b−e). The DFT model system is

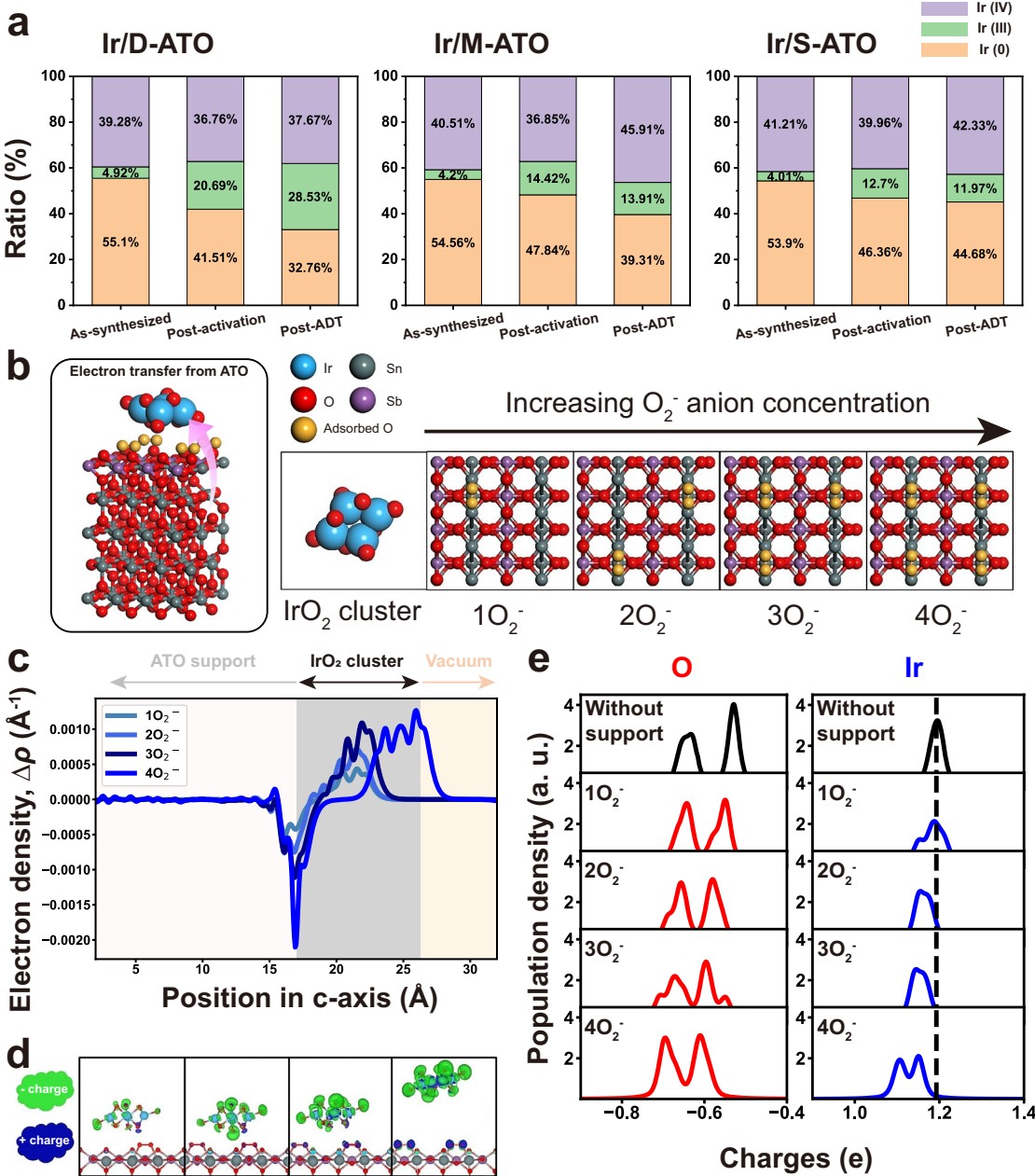

**Fig. 4 | Charge transfer interaction between the Ir catalyst and ATO support with varying amounts of EER. a** Characterization of the oxidation states of Ir in Ir/D-ATO, Ir/M-ATO, and Ir/S-ATO for each sample (as-synthesized, post-activation, post-ADT) by XPS. **b–e** DFT calculations of $O_2^-$ anion-mediated electron transfer from ATO support to Ir catalyst. **b** DFT model descriptions. $O_2^-$ concentrations (1 $O_2^-$ to 4 $O_2^-$) were varied to simulate the experimental systems of Ir/D-ATO, Ir/M-ATO, and Ir/S-ATO. **c** $O_2$ anion-induced electron transfer plotted along the *c*-axis, or $\Delta\rho = \rho_{total} - [\rho_{EER\text{-}contained\ support} + \rho_{catalyst}]$. **d** 3D visualizations of the charge transfer for the four model systems. **e** Population densities for atoms in the $Ir_4O_8$ cluster with varying $O_2^-$ concentrations. The cases termed 'without support' (the first row) refer solely to the $Ir_4O_8$ cluster. Note that a. u. represents arbitrary units.

composed of an $Ir_4O_8$ cluster and an underlying EER-contained ATO slab. The ATO slabs are (110) surface-exposed and contain 8 wt% antimony, which is close to the experimentally fabricated value (10 wt%). Within the EER, $O_2^-$ anion concentrations were controlled to simulate the differences among experimental Ir/D-ATO, Ir/M-ATO, and Ir/S-ATO. In Fig. 4c, d, where the electron transfer behaviors were visualized along the *c*-axis, the increase of $O_2^-$ concentration in EER indeed boosts the electron transfer from the ATO slab to the $Ir_4O_8$ cluster. This validates our hypothesis that the support incorporating the EER can modulate the strength of support-catalyst interactions.

Next, a Bader charge analysis in DFT modeling was performed to identify the charge states of Ir species in the catalysts ($Ir_4O_8$). The

donated electrons from the EER-contained ATO slabs would alter the charge states of each Ir and O atom, as shown in the population density plots in Fig. 4e. EER donates electrons to the $Ir_4O_8$ cluster, which results in decreases in the charge states of both Ir and O species. The increase of $O_2^-$ anion concentration causes a decrease of Ir charge states (the peaks moving to the left), which adequately explains the experimental observation that the relatively lower charge states of Ir (metallic Ir and Ir(III)) were the most extensively found in the Ir/D-ATO sample, compared to Ir/M-ATO and Ir/S-ATO. These DFT analyses reveal that the larger amount of Ir(III) in Ir/D-ATO results from the increased $O_2^-$ anion concentrations in the EER-contained ATO supports. Importantly, note that we also performed DFT calculations

using neutral oxygen molecule ($O_2$), instead of anions ($O_2^-$), and the variations of $O_2$ concentration had no effects on both electron transfer and Ir charge states (Supplementary Fig. 22). This supports the existence of $O_2^-$ species in our experimental samples.

## Single-cell performance in PEMWE

Furthermore, we demonstrated the superior catalytic performance of Ir/D-ATO in a PEMWE cell through comparison with the commercial Ir black catalyst and $TiO_2$-supported Ir catalyst (Ir/$TiO_2$) (Supplementary Fig. 24). The anode catalyst layers with the commercial catalysts were prepared with two Ir loadings, one with similar Ir loading of Ir/D-ATO ($10\ \mu g_{Ir}\ cm^{-2}$) and another with a much higher Ir loading ($500\ \mu g_{Ir}\ cm^{-2}$) (Supplementary Figs. 27 and 28). The higher Ir loading value was set to verify the formation of continuous catalyst layers[51]. Despite the ultra-low Ir loading ($7.2\ \mu g_{Ir}\ cm^{-2}$) of Ir/D-ATO, the catalytic performance of Ir/D-ATO far exceeded those of the Ir/$TiO_2$ and Ir black samples (Fig. 5a), and the mass activity of Ir/D-ATO was 55 and 75 times higher than those of high-loaded Ir/$TiO_2$ and Ir black at 1.6 V, respectively (Fig. 5b). In addition, the PEMWE single cell at Ir loading of $47.5\ \mu g_{Ir}$

$cm^{-2}$ operated for over 250 h at $1\ A\ cm^{-2}$ (Fig. 5e) and 500 h at $0.5\ A\ cm^{-2}$ (Supplementary Fig. 29) current densities without any significant degradation of cell voltage, with degradation rates of only 0.624 mV/h and 0.184 mV/h, respectively.

In particular, at low Ir loading, a uniformly connected 3D nanostructure of Ir/D-ATO (Supplementary Fig. 25) is desirable in that forming of a homogeneous and continuous catalyst layer is critical to good catalytic performance. As shown in Supplementary Fig. 26, the commercial Ir black and Ir/$TiO_2$ are composed of relatively larger nanoparticles (>100 nm), resulting in discontinuous and non-uniform catalyst layers at low Ir loadings (Supplementary Figs. 27a and 28a); it is highly likely that many small segments in the catalyst layer are not connected to the Ti porous transport layer (PTL) with a porosity of 60% (Supplementary Fig. 30). Consequently, the charge transfer resistance of the low-loaded Ir black and Ir/$TiO_2$ is considerably higher than that of the Ir/D-ATO (Fig. 5c), resulting in inferior catalytic performance. On the other hand, the high-loaded catalyst layers of the Ir black and Ir/$TiO_2$ are sufficiently continuous (Supplementary Figs. 27b and 28b) and demonstrate much higher catalytic performance than their

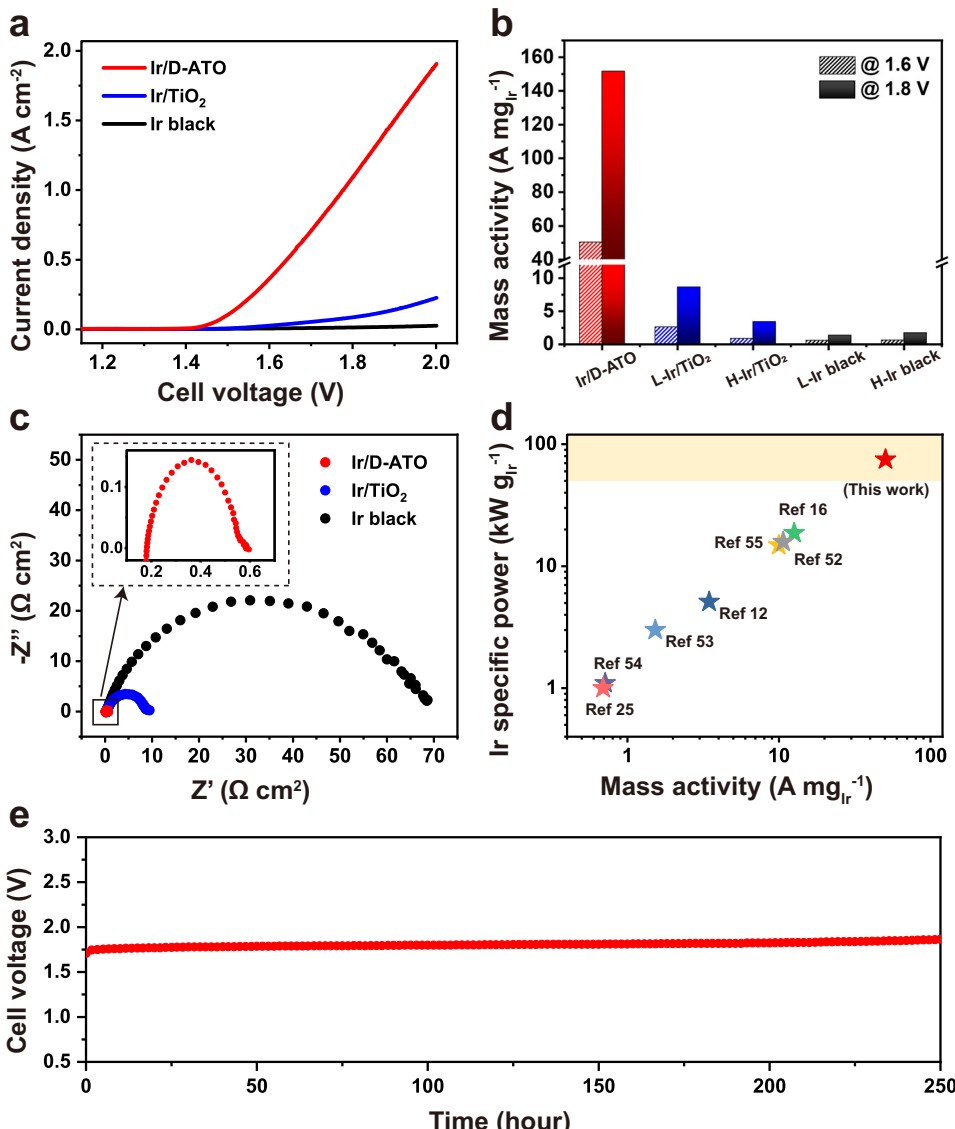

**Fig. 5 | Electrochemical measurements of Ir/D-ATO, Ir/TiO₂, and Ir black in PEMWE cell fed with deionized water. a** Polarization curves at low Ir loadings of $7.2\ \mu g_{Ir}\ cm^{-2}$ for Ir/D-ATO and $10\ \mu g_{Ir}\ cm^{-2}$ for Ir/$TiO_2$ and Ir black, **b** mass activity comparison estimated at 1.6 and 1.8 V (L-: low loading, $10\ \mu g_{Ir}\ cm^{-2}$, H-: high loading,

$500\ \mu g_{Ir}\ cm^{-2}$), **c** electrochemical impedance spectroscopy data measured at 1.5 V, **d** Ir-specific power vs. mass activity at 1.6 V plot with previous studies, and **e** chronopotentiometry data for stability test of Ir/D-ATO ($47.5\ \mu g_{Ir}\ cm^{-2}$) in PEMWE cell at $1.0\ A\ cm^{-2}$.

corresponding low-loaded catalyst layers (Supplementary Fig. 31a). This is attributable to the reduced charge transfer resistance, as evidenced by the Nyquist plots of the highly-loaded catalyst layers, which are not notably different from that of Ir/D-ATO (Supplementary Fig. 31b). Nevertheless, the Ir/D-ATO showed significantly higher mass activity than the high-loaded Ir black and Ir/TiO$_2$ samples; in addition to the favourable effect of the EER from D-ATO, the highly dispersed Ir nanoparticles on the ordered ATO architecture guarantee a considerably larger ECSA than the randomly packed Ir microparticles. Benefiting from the aforementioned features, the Ir/D-ATO exhibits the highest Ir-specific power of 74.8 kW g$^{-1}$ at PEMWE efficiency of ≈87%$_{HHV}$ (corresponding to a cell voltage of 1.6 V) compared to the previously reported Ir-based electrocatalysts, which were tested in a PEMWE cell (Fig. 5d)[12,16,25,52–55]. The Ir-specific power figures were calculated by the following equation:

$$\text{Ir specific power} \left(\text{kW g}^{-1}\right) = \frac{i \times HHV}{2F \times l} \times 1000 \, (\text{mg/g}) \quad (1)$$

where $i$ is the current density (A cm$^{-2}$), $HHV$ is the higher heating value of H$_2$ (286 kJ mol$^{-1}$), F is the Faraday constant (96485 C mol$^{-1}$), and $l$ is the Ir loading (mg cm$^{-2}$). The high Ir-specific power of Ir/D-ATO shows the potential to further extend its composition and patterned nanostructure in practical circumstances. In addition, the actual electrode size achievable with the nanotransfer printing technique that we used depends on the size of the master template wafer, and scalability up to 8 inches was already demonstrated in our previous study[56]. Overall, it may be promising in gigawatt-scale H$_2$ production since the necessary Ir-specific power for such scale was predicted to be 50–100 kW g$^{-1}$ depending on several conditions[57], which was unprecedented until the present work.

## Discussion

In summary, we successfully designed and fabricated the EER-contained metal oxide supports as a unique strategy to simultaneously improve the catalytic activity and the durability of Ir-based electrocatalysts based on enhanced charge donation by the EER. As a result, the highest values for both mass activity and Ir-specific power, which were estimated in a single-cell PEMWE, among Ir-based electrocatalysts reported to date were achieved. Moreover, the PEMWE incorporating the EER-supported Ir catalyst exhibited high durability over 250 h at 1.0 A cm$^{-2}$ condition despite remarkably low loading of Ir. We provided evidence that the degree of charge transfer of metal oxide supports, which is a key parameter in improving the OER performance, can be artificially controlled by engineering the density of EER, as revealed by our theoretical and experimental studies. More specifically, we confirmed that a higher density of EER contributes to the maintenance of a higher Ir(III) to Ir(IV) ratio during ADT. Our findings overcome the limited charge transfer properties of typical metal oxides by forming the EER at the surface as an excellent promotor of charge transfer, suggesting new design rules to develop metal oxide support materials for higher performance OER electrocatalysts. We believe that our strategy can be extended to various metal oxide applications where the overall performance is closely related to charge transfer capability, including fuel cells, optoelectronics, and photovoltaics.

## Methods

### Fabrication of ATO nanostructure supported Ir catalysts

Figure 2b schematically illustrates the fabrication process of ATO nanostructure supported Ir catalysts (Ir/ATOs) through S-nTP, a previously reported technique[41]. A line-patterned master template pretreated by a polydimethylsiloxane (PDMS, Polymer Source Inc.) brush with a low surface energy was prepared. On top of the line-patterned master template, a polymethylmethacrylate (PMMA, Sigma-Aldrich

Inc.) solution was spin-coated to form a polymer replica. Polyimide adhesive tape (PI, 3 M Inc.) was then attached to the surface of the polymer replica to peel off the polymer replica from the master template. Both metals (Ir, Sb, iTASCO) and metal oxide (SnO$_2$, iTASCO) were deposited sequentially through oblique-angle deposition (tilt angle: 80°) to form a discrete nanowire array on the PMMA replica using an e-beam evaporator. The EER content of the deposited SnO$_x$ nanowires could be adjusted by controlling the start time of e-beam evaporation through the shutter. The start time was specified by the XPS depth profile (Fig. 1d) of the e-beam deposited SnO$_x$ thin film. For dense EER SnO$_x$ nanowires, the shutter was opened immediately after the e-beam was turned on. The shutter was opened for sparse EER SnO$_x$ nanowires after 20 nm of SnO$_2$ was deposited according to the thickness monitor. After deposition of all the components, the discrete nanowire array was transfer-printed onto a Cu foil and then the PMMA replica was removed by toluene immersion. By repeating the transfer-printing of the deposited nanowire array in the perpendicular direction of the previous array sequentially, multilayer stacked 3-dimensional nanostructures were formed. Finally, the resulting product was annealed at 700 °C for 2 h in Ar conditions to form ATO and spherical shape Ir nanoparticles on the surface of ATO (Supplementary Fig. 10). To transfer the Ir/ATOs onto a glassy carbon working electrode for half-cell measurement or a Nafion membrane for single-cell measurement, the Cu foil was wet-etched with a 0.1 M ammonium persulfate solution (Sigma-Aldrich Inc.). After etching the Cu substrate, the Ir/ATOs floating in the ammonium persulfate solution were transferred to a specified substrate for electrochemical measurements. In the case of Ir/CNW, all the fabrication processes described above were equally applied except for depositing graphite carbon as a support material.

### Characterization of materials

Structural and physiochemical characterizations of the Ir/ATOs were carried out with various methods including SEM, TEM, Raman spectrometry, XRD, XPS, and ICP-MS. The morphology of the Ir/ATOs was investigated by field emission SEM (Hitachi, S-4800) with an acceleration voltage of 10 kV and field emission TEM (FEI, Tecnai G2 F30 S-Twin) operated at an acceleration voltage of 300 kV. HAADF-STEM, SAED pattern, and EDX mapping images were acquired using a TEM (JEOL, JEM-2100F HR) operated at an acceleration voltage of 200 kV. Raman spectra were recorded using a dispersive Raman spectrometer (Horiba Jobin Yvon, ARAMIS) with a 514 nm Ar-ion laser. XRD (RIGAKU, SmartLab) measurements were conducted in θ–2θ scan mode using a Cu $K_{\alpha1}$ incident beam to analyze the crystal information of the Ir-ATOs. XPS (Thermo VG Scientific, K-Alpha) measurements were performed to investigate the chemical compositions and oxidation states of the Ir/ATOs. The C 1$s$ peak at 284.8 eV was used as the reference for the calibration of binding energies. ICP-MS (Agilent, ICP-MS 7700S) experiments were conducted at least five times to measure the total amount of Ir loading in each catalyst.

### Electrochemical half-cell tests

The electrochemical characterizations of Ir-ATOs and other references such as Ir/C were carried out in a rotating disk electrode system with a potentiostat (Garmry, Interface 1010E) at room temperature. A 0.05 M H$_2$SO$_4$ solution (98%, Sigma-Aldrich Inc.) was used as an electrolyte. A conventional three-electrode half-cell setup consisting of glassy carbon working electrode (Pine, 0.196 cm$^2$), a platinum mesh counter electrode, and an Ag/AgCl reference electrode was used. The rotation speed of the working electrode was controlled with a modulated speed rotator (Pine Inc.). All of the potentials indicated in electrochemical data were converted to the RHE scale by equation below:

$$E_{\text{RHE}} = E_{\text{Ag/AgCl}} + 0.059 \times \text{pH} + E^{\text{o}}_{\text{Ag/AgCl}} \quad (2)$$

where $E°_{Ag/AgCl}$ is 0.197 V at room temperature. 100% iR-correction was conducted to compensate for ohmic drop in the electrolyte. A mixture of 3.5 mg of Ir black commercial catalyst (Premetek), 2.0 mL of deionized (DI) water, 8.0 mL of isopropyl alcohol, and 40 µL of Nafion solution (5 wt%, Sigma-Aldrich Inc.) was prepared as a catalyst ink. After sonication of the prepared ink solution for about 30 min, 10 µL of catalyst ink was dropped onto a working electrode, where the corresponding amount of Ir loading was about 17.8 µg cm$^{-2}$. The same processes were applied to the Ir/C (20 wt%, Premetek) commercial catalysts.

First, cyclic voltammetry (CV) tests were conducted in a range of 0.05 V to 1.4 V at a scan rate of 200 mV s$^{-1}$ in Ar-saturated 0.05 M H$_2$SO$_4$ electrolyte until the entire surface of the Ir catalyst was oxidized to a hydrated Ir oxide. The formation of hydrated Ir oxide can be confirmed through the absence of further change in the CV curves or the decrease of the HUPD peak intensity. After activation, linear sweep voltammetry (LSV) tests were carried out between 1.2 V and 1.7 V at a scan rate of 5 mV s$^{-1}$ under a working electrode rotation of 1600 rpm. Electro-chemical impedance spectroscopy (EIS) was conducted at 1.55 V$_{RHE}$ from 1 Hz to 100 kHz. The intercept with x-axis at a high frequency region in the Nyquist plot (34–37 Ω) was used as electrolyte resistance for iR-correction. A long-term stability test of each catalyst was accomplished through chronopotentiometry (CP), where the potential value was measured to obtain constant current densities of 1 and 10 mA cm$^{-2}$ with a rotation speed of 1600 rpm. At 1 and 10 mA cm$^{-2}$ conditions, a glassy carbon working electrode and Ti plate were used as substrates, respectively. In addition, an accelerated durability test (ADT) was conducted via potential-cycling between 1.2 V and 1.6 V at a scan rate of 50 mV s$^{-1}$. The electrochemically active surface area (ECSA) of each catalyst was calculated by the CO stripping method. The current peak over the range of 0.7 V to 1.15 V corresponds to the deso-rption peak of CO molecules on the Ir surface. The peak area was calculated and normalized by the specific charge value (420 µC cm$^{-2}$$_{Ir}$) to give the ECSA.

## DFT calculations

DFT calculations were carried out using the plane-wave-basis Vienna ab initio simulation package (VASP) code with an energy cutoff of 400 eV[58,59]. The projector-augmented wave method was adopted to treat core and valence electrons[60]. The generalized gradient approx-imation was used to describe the exchange-correlational interactions with the Perdew-Burke-Ernzerhof functional[61]. To calculate the simu-lation systems with the several EER candidates in Fig. 1, we modeled the (110) ATO surfaces of periodically repeated 2 × 2 supercells with three trilayers and a vacuum width of 15 Å. Here, the Brillouin zone was sampled with a 2 × 2 × 1 Monkhorst−Pack k-point mesh. To describe the O$_2^-$ anion as an EER in more detail, we modeled a larger system in the calculations of Fig. 4 than that those of Fig. 1. The ATO support was described as 2 × 4 supercell in the lateral direction and a five-trilayer-thick ATO (110) surface with a vacuum width of 20 Å. The most pre-ferred adsorption configuration of O$_2^-$ anion on the ATO (110) surface is side-on adsorption between two Sn atoms (Supplementary Fig. 23). For the larger supercell, Brillouin-zone integrations were performed using Monkhorst−Pack k-point samplings of 1 × 2 × 1. The geometry was fully relaxed until the maximum Hellmann−Feynman forces were less than 0.02 eV/Å, and the electronic structures were relaxed with a convergence criterion of 10$^{-5}$ eV. A Bader charge analysis was per-formed to investigate the charge states of Ir and O atoms in the Ir$_4$O$_8$ cluster[62].

## Membrane electrode assembly (MEA) preparation and single-cell measurements

The Ir/D-ATO catalyst was loaded on a Nafion 212 membrane (1 cm$^2$ active area) as the anode catalyst, and a Pt/C catalyst (46.3 wt%, Tanaka) was used as the cathode catalyst. For comparison, Ir black (Premetek) and Ir/TiO$_2$ (Elyst 75, Umicore) were used as the anode catalysts. Besides the Ir/D-ATO, a homogeneous catalyst ink was pre-pared by mixing each catalyst with 5 wt% Nafion solution and iso-propanol. A specific catalyst ink was sprayed directly onto the Nafion 212 membrane (1 cm$^2$ active area). For the cathode, 0.1 mg$_{Pt}$ cm$^{-2}$ (Pt/C) was loaded with 25 wt% Nafion content, and 39BB (Sigracet) was used as a gas diffusion layer. For the anode, 0.01 or 0.5 mg$_{Ir}$ cm$^{-2}$ (Ir black and Ir/TiO$_2$) was loaded with 10 wt% Nafion content, and Ti felt (Bekaert, 250 µm) was used as a porous transport layer. Single-cell measurements were conducted with a potentiostat (Metrohm Autolab PGSTAT302N) equipped with a 10 A current booster. The cell was operated at 80 °C under ambient pressure, and deionized water was fed to the anode side at a flow rate of 15 mL min$^{-1}$. The EIS was con-ducted at 1.5 V from 10$^{-1}$ to 10$^5$ Hz with a potential amplitude of 10 mV.

## Reporting summary

Further information on research design is available in the Nature Portfolio Reporting Summary linked to this article.

## Data availability

All the data that support the findings of this study are available from the corresponding authors upon reasonable request.

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

## Acknowledgements

This work was supported by Korea Institute of Energy Technology Evaluation and Planning (KETEP) grant funded by the Korean government (MOTIE) (No. 20214000000650, Energy Innovation Research Center for Fuel Cell Technology). This work was supported by the program of Future Hydrogen Original Technology Development through the National Research Foundation (NRF) of Korea funded by the Ministry of Science and ICT (NRF-2021M3I3A1082879). This work was also supported by the National Research Foundation (NRF) of Korea funded by the Ministry of Science and ICT (NRF-2022M3H4A7046278).

## Author contributions

G.R.L. and Y.S.J. conceived the project. G.R.L. conducted the fabrication process, prepared samples, and carried out structural and chemical characterizations. G.R.L. and J.K. performed the electrochemical experiments and analyzed the data. D.H. and D.K. performed the DFT calculations and theoretical analyses. G.R.L., Y.J.K., H.J., H.J.H., and C.H. optimized the preparation process for the analysis. G.R.L. and Y.S.J. led the discussions and wrote the manuscript with input from all authors. Y.S.J., J.Y.K., and D.K. supervised the project. All authors discussed the results and contributed to the manuscript.

## Competing interests

The authors declare no competing interests.
