## [Peer Review File · Nature Communications]

Efficient and Sustainable Water Electrolysis Achieved by Excess Electron Reservoir Enabling Charge Replenishment to CatalystsREVIEWER COMMENTS

Reviewer #1 (Remarks to the Author):

This manuscript by Lee et al. reports a study on the novel support modification strategy of introducing an excess electron reservoir (EER) on metal oxide supports and the oxygen evolution performance of obtained EER-supported Ir catalyst (Ir/D-ATO). The authors claimed that the EER-decorated ATO support could serve as an electron-donating layer to promote the charge transfer to IrO_x and, as a result, stabilize active-Ir(III) species without structural deformation. By combining theoretical and experimental analyses, the authors believed that the EER on the support contributes to the maintenance of active-Ir(III), which plays an important role in the improved OER performance of Ir/D-ATO. These results are likely to be interesting in the research field of sustainable water electrolysis. However, some conclusions are not fully supported by the present experiment results. Furthermore, more experimental results are needed to verify the stability of Ir/D-ATO. Therefore, in its present form, I consider that the manuscript cannot meet the high standards of Nature Communications. There are a number of critical issues that need to be resolved.

1. The definitions of S-ATO, M-ATO and D-ATO were not clear in terms of preparation methods and characterizations. The specific intensity or ratio change in XPS analysis should be presented.
2. The long-term stability of the Ir/D-ATO was assessed only at a current density of 100 mA cm⁻² in PEMWE. Can the catalyst electrode survive >1 A/cm² or >2 V operating conditions, which are current state-of-the-art performance parameters for PEMWE [Angew. Chem. Int. Ed. 2023, 202216645; Small Methods 2022, 6, 2201130]? How long do the electrodes last under more realistic high current/voltage conditions?
3. The accelerated durability test (ADT) was conducted via potential-cycling. Is the ratio of Ir(III) species maintained during the chronopotentiometry measurement? What about the structure evolution of the Ir/D-ATO after the stability testing?
4. Could the multilayer stacked 3D nanostructures of catalyst be damaged during the etching process of Cu foil? How did the authors obtain the amount of Ir loading transferred to the glassy carbon electrode or Nafion membrane?
5. What caused the ECSA error for Ir/CNW to be significantly larger than for other catalysts in Fig. 3c?
6. The authors claimed that the increase of O²⁻ anion concentration caused a decrease of Ir charge states (the peaks moving to the left in Fig. 4e). Why did the O charge states also decrease simultaneously?
7. There are some expression errors in this manuscript, such as the fraction of each component of the Ir/D-ATO in page 7. The authors should carefully check the full text and correct them. Besides, please make consistency with the nomenclature, specifically Ir black or IrB, in all the figures.

Reviewer #2 (Remarks to the Author):

Response to authors: Accept with Major revisions

The manuscript by Lee and co-workers explores the preparation of supported IrO_x catalyst onto an antimony-doped tin oxide with an excess of surface superoxide moieties to stabilize the Ir³⁺ active phase under oxygen evolution reaction conditions. Besides physicochemical characterization (STEM, XPS, XRD), the authors performed long-term durability testing on a PEMWE cell yielding promising results in terms of stability. Despite the aforementioned, there are several scientific questions to be addressed before consideration for publishing in Nature Communications:

- 1) Despite the thorough XPS spectra deconvolution, the authors have disregarded the well-known asymmetric nature of the Ir 4f spectra. This, firstly postulated by Doniach and Šunjić (J. Phys. C.: Solid State Phys., 1970, 3, 285-291), has been adopted by the XPS community by employing Functional Lorentzian (LF) lineshapes. This can result in an underestimation of the main peak contribution (Ir³⁺ and Ir⁴⁺) to the overall spectral signal. In addition, the contribution of the Ir 5p ¹/₂ peak should be included in the peak deconvolution, generally found in the 64-65 eV range. Such

contribution will unambiguously affect the Ir³⁺/Ir⁴⁺ analysis, as the Ir³⁺ doublet lies in such window. The authors should correct this by using the peak deconvolution lineshapes suggested by Freakley et al. (Surf. Interface Anal. 2017, 49, 794–799) and Pfeifer et al. (Surf. Interface Anal. 2016, 48, 261–273) to interpret the data. Authors should cite the following relevant works of such topic: Energy Environ. Sci., 2013, 6, 3756–3764; Phys. Chem. Chem. Phys., 2016, 18, 2292; ACS Catal. 2021, 11, 15, 9300–9316.

2) Following up with the XPS characterization, the authors ascribe a Sn satellite at higher binding energies upon increasing superoxide contents on ATO. Could the authors provide further insights on this? I would have anticipated, unless a total formal charge is present in ATO, that either Sb or Sn are present in a lower oxidation state. However, previous XPS studies would suggest the aforementioned satellite would correspond indeed to Sn⁴⁺ and the Sn⁴⁺ peak to that of Sn²⁺ (Kwoka et al., Thin Solid Films 2005, 490, 36–42) although it is hard to tell given that there is no information on the exact deconvoluted peak binding energies, which should be provided in the revised version of the manuscript. In addition to this, can the authors explain why are there no differences in Ir chemical states in pristine samples with varied EER contents?

3) Can the authors report the XPS spectra of Ir/CNW? This would be the easiest approach to prove the CSI effect of ATO versus an alternative support with the same Ir synthetic method.

4) In order to report electrochemically active surface areas, the authors employ here CO stripping. The authors should justify why CO stripping was employed instead of more consolidated methods in the literature for Ir-based catalysts such as Hg UPD (Alia et al., Journal of The Electrochemical Society, 163 (11) F3051-F3056 (2016)), Zn cation adsorption (Zhao et al., Journal of The Electrochemical Society, 162 (12) F1292-F1298 (2015)) or adsorption capacitance measurements (Watzel et al., ACS Catal. 2019, 9, 9222–9230). CO stripping is based upon substrate-specific interactions, and there is no proof that CO adsorption selectively takes place only on top of Ir and not on Sb or Sn.

5) Regarding the authors statement on Page 8 'In general, it is well known that the interaction between the carbon support and catalysts is too weak to augment the intrinsic activity of catalysts¹⁷', it is well known in the OER community that carbon is not an adequate support for acidic oxygen evolution electrocatalysis as it undergoes corrosion at anodic potentials (Yi et al., Catalysis Today 2017, 295, 32–40; Geiger et al., ChemSusChem 2017, 10, 4140–4143; Browne et al. J. Mater. Chem. A 2018, 6, 14162–14169; Edginton et al., ACS Appl. Energy Mater. 2022, 5, 10, 12206–12218). Thus, the authors should take this into account when evaluating the electrochemical data, particularly relevant in the poor stability of Ir/C (Supplementary Fig. 13), which surely stems from carbon corrosion and subsequent Ir particle detachment from the working electrode.

6) Electrocatalyst assessment based purely on potentiostatic holds are well-known to provide misleading metrics in terms of electrocatalyst stability as catalyst layer quality, active site blockage by microbubbles or surface oxidation state changes ultimately affect the experimental results. The following works discuss these issues at length, which should be discussed and cited in the manuscript: Kibsgaard et al., Nat. Energy 2019, 4, 430–433; R. Frydendal et al. ChemElectroChem 2014, 1, 2075; Lazaridis et al., Nat. Catal. 2022, 5, 363–373; Ehelebe et al., Curr. Opin. Electrochem. 2021, 29, 100832. The authors should evaluate the catalyst stability by quantifying dissolved amounts of Ir/Sb/Sn in the acidic electrolyte with ICP-MS to unambiguously prove that indeed the catalyst-support interactions stabilize the Ir³⁺ active sites.

7) XPS characterization of Sb and Sn should be provided after ADT testing to showcase if there are any significant oxidation state changes at the support as well as any presence of the superoxides obtained during oxygen-rich evaporative deposition conditions.

8) Single-cell performance in PEMWE: catalyst layer imaging for Ir/D-ATO should be provided to evaluate the quality of the spraycoating. In addition, the authors should have employed a commercial supported Ir catalyst such as Elyst75 (<https://fcs.umicore.com/en/fuel-cells/products/elyst-ir75-0480>) to provide a more realistic benchmarking instead of an unsupported Ir catalyst which clearly yielded inhomogenous catalyst layers at low loadings. Regarding the effect of Ir loadings in PEMWE performance, the authors should cite the work of Gasteiger's group (Bernt et al., J. Electrochem. Soc. 2018, 165, 5, F305-F314).

Minor points:

- 1) The authors fail to cite relevant references concerning the degradation mechanisms of Ir:
Kasian et al., *Angew. Chem. Int. Ed.* 2018, 57, 2488-2491
Kasian et al., *Energy Environ. Sci.*, 2019, 12, 3548-3555
Scott et al., *Energy Environ. Sci.*, 2022,15, 1988-2001
- 2) The statement on page 3, second paragraph 'Alternatively, sustainably retaining active-Ir(III) species is considered one of the most desired strategies to overcome the fundamental trade-off relationship between activity and durability' should be backed up by relevant references.
- 3) The section labelled as Discussion corresponds to the Conclusions of the manuscript. This should be amended in the revised version of the manuscript.

Reviewer #3 (Remarks to the Author):

The authors present an anode catalyst for PEM water electrolysis consisting of IrOx nanoparticles decorated on an ATO support. The ATO was grown by e-beam evaporation-deposition, and the content of O₂⁻ species could be controlled by variation in time, forming what the authors call a an excess electron reservoir (EER, O₂⁻). These supported catalysts were characterized and the conclusions are supported by DFT calculations and screening. The catalyst layer was then fabricated by solvent-assisted nanotransfer printing method allowing a high surface area 3D structure, which was tested in half-cell and single full cell. The catalyst showed remarkable Ir-specific power at 1.6 V using a very low Ir loading.

This work is interesting for the development of Ir catalysts supported on metal oxides and for the presented fabrication method of the anode catalyst layer for PEMWE. I recommend publications after revision.

Comments:

Abstract, line 26: "an excess electron reservoir (EER, O₂⁻)decorated on antimony-doped tin oxide"
When I first read the abstract it was not clear to me what the EER exactly is. I think this has to be clarified in the abstract. Also the use of "decorated on" is confusing, since it seems to me more as "incorporated in" as used also later in the text is more appropriate. My understanding is that this EER is just a part of the ATO film that presents a specific oxygen species (O₂⁻), and is created during the fabrication of the ATO layer, not exactly a surface decoration, which might confuse the reader implying that there is an heterostructure on top of the ATO.

Abstract, line 33: "remarkable potential for realizing gigawatt-scale H₂ production" and page 12, line 337-338. I have a question regarding scalability, could the authors state in the text (not necessary in the abstract) what is the actual electrode size achievable with the nanotransfer technology and what are the promises of this interesting method.

Page 6, line 131: Raman is used here with the "model" EER-incorporated SnOx for a direct evidence of the O₂⁻, but later in the manuscript XPS is chosen. Could the author comments about the challenge of Raman characterization with nanotransferred sample? Or what is the reason?

Page 6, Line 133: "a proper initiation point". This expression is not clear.

Page 6, line 138 and Supp. Figure 5: In the caption of Supp Figure 5, it is not explained which of the species is the O₂⁻.

Page 8, line 207: "highest mass activity value...(Supp. Table 2)". The sentence is correct, but it is noticeable how the Ir loading is much lower than the other compared catalysts, and Ir mass is in the

denominator of mass activity. I think the authors should also compare specific activity in the table, for a more fair comparisons.

Page 9, line 225: "by holding the potential to obtain constant current densities", maybe a typo here...

Page 9, line 233: "with such a high current density" 10 mA cm⁻² is not high, and the current chosen for the stability test of 1 mA cm⁻² shown in Figure 3b is also very low. Could the authors justify their choices?

Page 9, line 251: "The tafel slope of Ir/D-ATO (49.05 mV dec⁻¹, Fig 3e) is significantly lower compared to Ir/M-ATO (51.40 mV dec⁻¹)..." . I would argue that the difference of 2 mV dec⁻¹ is not a big difference. If the authors want to use the term "significant" they should show uncertainties from reproductions. Otherwise I would suggest to simply rephrase the sentence. With S-ATO is ok, there the difference is 10 mV dec⁻¹.

Figure 1d: In the caption it would be useful to specify how the three regions are identified, which I guess it is from the fitting of the oxygen XPS peak. For example, otherwise it is unclear why the limit between medium and sparse is at ~150 etch seconds. Maybe the Si substrate region can also be indicated, but this last is a minor change.

Figure 2b): This figure panel is not clear to me. For example, the supporting figure S9a is much clearer. In my opinion, the issue with Figure 2b is the lack of labels and the fact that the single steps are not described by a short underneath text. Also the 3D structure with the alternately stacked layers is not fully shown, in contrast to figure s9a.

Figure 2c and caption: In the caption the scale bar for 2c is described in the text referred to d), after point d).

■ Point-by-point responses to the reviewer's comments.

We provide point-by-point responses to the reviewer's in-depth comments, and the updated text in the revised manuscript is marked in blue.

Reviewer #1

[General remarks] *This manuscript by Lee et al. reports a study on the novel support modification strategy of introducing an excess electron reservoir (EER) on metal oxide supports and the oxygen evolution performance of obtained EER-supported Ir catalyst (Ir/D-ATO). The authors claimed that the EER-decorated ATO support could serve as an electron-donating layer to promote the charge transfer to IrOx and, as a result, stabilize active-Ir(III) species without structural deformation. By combining theoretical and experimental analyses, the authors believed that the EER on the support contributes to the maintenance of active-Ir(III), which plays an important role in the improved OER performance of Ir/D-ATO. These results are likely to be interesting in the research field of sustainable water electrolysis. However, some conclusions are not fully supported by the present experiment results. Furthermore, more experimental results are needed to verify the stability of Ir/D-ATO. Therefore, in its present form, I consider that the manuscript cannot meet the high standards of Nature Communications. There are a number of critical issues that need to be resolved.*

[General response] We appreciate the reviewer's positive evaluation and insightful comments that enabled us to greatly improve our manuscript. The reviewer pointed out several valid concerns, especially in terms of the stability of Ir/D-ATO. Therefore, we have performed additional experiments and made significant corrections in the revised manuscript, and the following are our point-by-point responses.

[Comment 1] *The definitions of S-ATO, M-ATO and D-ATO were not clear in terms of preparation methods and characterizations. The specific intensity or ratio change in XPS analysis should be presented.*

[Response 1] We appreciate the reviewer's detailed comment regarding the clear definitions for distinguishing between S-ATO, M-ATO, and D-ATO. We agree that they need to be distinguished quantitatively using XPS data. In response, we prepared two criteria: (1) O/Sn area ratio and (2) $\text{Sn}^{\text{Satellite}}/\text{Sn}^{\text{IV}}$ ratio, which can be both quantified by XPS analysis. First, we calculated the O/Sn ratio based on the XPS peak intensity. The density of EER can be correlated with the O/Sn ratio because the charged-oxygen group (especially, O_2^-) is selected as EER in this study. As shown in **Supplementary Fig. 6a**, D-ATO shows the highest O/Sn ratio of 0.94, which is larger than 0.569 for M-ATO and 0.401 for S-ATO.

Second, the $\text{Sn}^{\text{Satellite}}/\text{Sn}^{\text{IV}}$ ratio, which can be obtained through the deconvolution of the Sn 3d XPS peak, is another useful indicator. As shown in **Supplementary Fig. 5**, the Sn 3d peaks can be deconvoluted into Sn^{IV} and $\text{Sn}^{\text{Satellite}}$ peaks, which are assigned to 486.6 eV¹ and a higher binding energy than 486.6 eV, respectively. Compared to the bulk sample, where only the 486.6 eV peak appears, the S-ATO, M-ATO, and D-ATO samples with different EER contents show both the Sn^{IV} and the $\text{Sn}^{\text{Satellite}}$ peaks. It can be seen in **Supplementary Fig. 6b** that the ratio of $\text{Sn}^{\text{Satellite}}$ to Sn^{IV} is highest for D-ATO and lowest for S-ATO. This is consistent with the prediction that, as the density of EER increases, it is more difficult to detach electrons from Sn, and thus, the portion of $\text{Sn}^{\text{Satellite}}$ increases. More detailed explanation about the binding energy of Sn containing EER is provided in **Reviewer #2 – [Response 2]**.

To summarize, both the O/Sn area ratio and the $\text{Sn}^{\text{Satellite}}/\text{Sn}^{\text{IV}}$ ratio can be used as indicators that can clearly distinguish S-ATO, M-ATO, and D-ATO. Accordingly, in response to the reviewer's comment, the main manuscript and the Supplementary Information were updated as follows.

Reference

1. Kwoka M, Ottaviano L, Passacantando M, Santucci S, Czempik G, Szuber J. XPS study of the surface chemistry of L-CVD SnO₂ thin films after oxidation. *Thin Solid Films* **490**, 36-42 (2005).

[Modification in the manuscript]

(Result, page 6) “The O/Sn and $\text{Sn}^{\text{Satellite}}/\text{Sn}^{\text{IV}}$ ratios can experimentally distinguish the S-ATO, M-ATO, and D-ATO samples. It can be seen that as more EER is incorporated, the intensity of the oxygen peak becomes higher compared to the Sn peak and the ratio of O_2^- to lattice oxygen is also higher, in comparison with the bulk sample. Indeed, D-ATO presents the highest O/Sn ratio of 0.94, which is larger than the values of 0.569 for M-ATO and 0.401 for S-ATO. In addition, it can be seen that as more EER is contained, the ratio of $\text{Sn}^{\text{Satellite}}/\text{Sn}^{\text{IV}}$ obtained through deconvolution of the Sn 3d spectra is higher (Supplementary Fig. 6). These results ~”

(Supplementary Information)

Supplementary Fig. 6. Criteria for the definitions of S-ATO, M-ATO, and D-ATO. a O/Sn area ratio and b $\text{Sn}^{\text{Satellite}}/\text{Sn}^{\text{IV}}$ ratio of the ATO with varying EER content. The O/Sn ratio was analyzed using the peak intensity of the O 1s and Sn 3d peaks. The $\text{Sn}^{\text{Satellite}}/\text{Sn}^{\text{IV}}$ ratio was calculated using the peak intensity of the Sn 3d peaks deconvoluted into $\text{Sn}^{\text{Satellite}}$ and Sn^{IV} .

Supplementary Fig. 5. X-ray photoelectron spectroscopy (XPS) spectra of bulk and EER-contained ATO nanowire arrays with varying density of EER. The binding energies of Sn 3d spectra (a, c, e, g) and O 1s spectra (b, d, f, h) of each sample, respectively.

[Comment 2] *The long-term stability of the Ir/D-ATO was assessed only at a current density of 100 mA cm⁻² in PEMWE. Can the catalyst electrode survive >1 A/cm² or >2 V operating conditions, which are current state-of-the-art performance parameters for PEMWE [Angew. Chem. Int. Ed. 2023, 202216645; Small Methods 2022, 6, 2201130]? How long do the electrodes last under more realistic high current/voltage conditions?*

[Response 2] We appreciate the reviewer's helpful comment and agree with the suggestion regarding the operating conditions for the long-term stability test. We conducted an additional stability test at a higher current density of 1 and 0.5 A cm⁻², and our Ir/D-ATO catalyst successfully maintained its initial performance with a marginal degradation for 250 h (0.624 mV/h) and 500 h (0.184 mV/h), respectively.

[Modification in the manuscript]

(Abstract, page 2) “When used in a polymer electrolyte membrane water electrolyzer, ~ and outstanding long-term stability for 250 h with a marginal degradation under a water-splitting current of 1 A cm⁻².”

(Main, page 4) “The Ir catalysts placed on EER-incorporated ATO demonstrated significantly enhanced mass activity, around 75 times higher than those of commercial Ir nanoparticle catalysts, and long-term stability over 250 h at 1.0 A cm⁻² current density (0.624 mV/h) in a PEMWE single cell.”

(Result, page 12) “In addition, the PEMWE single cell at Ir loading of 47.5 μg_{Ir} cm⁻² operated for over 250 h at 1 A cm⁻² (**Fig. 5e**) and 500 h at 0.5 A cm⁻² (**Supplementary Fig. 29**) current densities without any significant degradation of cell voltage, with degradation rates of only 0.624 mV/h and 0.184 mV/h, respectively.”

(Discussion, page 14) “Moreover, the PEMWE incorporating the EER-supported Ir catalyst exhibited high durability over 250 h at 1.0 A cm⁻² condition despite remarkably low loading of Ir.”

Fig. 5. Electrochemical measurements of Ir/D-ATO, Ir/TiO₂, and Ir black in PEMWE cell. a Polarization curves ~, and **e** chronopotentiometry data for stability test of Ir/D-ATO in PEMWE cell at 1.0 A cm⁻².

(Supplementary Information)

Supplementary Fig. 29. Electrochemical measurements of Ir/D-ATO in PEMWE single cell. Chronopotentiometry data for stability test of Ir/D-ATO in PEMWE single cell at 0.5 A cm⁻².

[Comment 3] *The accelerated durability test (ADT) was conducted via potential-cycling. Is the ratio of Ir(III) species maintained during the chronopotentiometry measurement? What about the structure evolution of the Ir/D-ATO after the stability testing?*

[Response 3] We appreciate the comment from the reviewer on the chemical composition change and structural evolution of the Ir/D-ATO after chronopotentiometry (CP) measurement. In response to this comment, we further conducted XPS and SEM characterizations after the CP measurement performed for the stability test. The CP measurement was conducted in parallel by monitoring the potential value to obtain a constant current density of 1 mA cm^{-2} during the same time as much as the accelerated durability test (ADT). The Ir 4f XPS spectra were deconvoluted to compare the ratio of metallic Ir, Ir(III), and Ir(IV) of the Ir/D-ATO attained in each durability test type: after ADT (**Fig. R1a**), and after CP (**Fig. R1b**). We confirmed that there is no significant difference in the chemical states of Ir in spite of the durability test type and the summary of the ratio is described in **Fig. R1c**. Especially, the similar proportion of Ir(III) species that originated from two different durability tests indicates that the effects of the EER on the oxidation states of IrO_x operate consistently regardless of the method used. Finally, the structure evolution of the Ir/D-ATO after the stability test was confirmed through SEM analysis (**Fig. R1d**), and carbon paper was used as a conductive substrate. The initial morphology of Ir/D-ATO which shows three-dimensional nanostructures composed of multilayer-stacked nanowire arrays was well maintained after the stability test without any critical structural deformation or collapse. Thus, considering these results, we could conclude that the enhanced durability of Ir/D-ATO regardless of the test type mainly originated from the charge transfer driven by EER and the secure retention of three-dimensional nanostructures.

Fig. R1. Physicochemical Characteristics of Ir/D-ATO after durability tests with both ADT and chronopotentiometry measurements. X-ray photoelectron spectroscopy (XPS) spectra of the Ir/D-ATO **a** after ADT and **b** after the chronopotentiometry measurement. **c** Characterization of the oxidation states of Ir in Ir/D-ATO measured after each type of stability tests. **d** SEM images of Ir/D-ATO after stability test with 1 μm scale bar.

[Comment 4] *Could the multilayer stacked 3D nanostructures of catalyst be damaged during the etching process of Cu foil? How did the authors obtain the amount of Ir loading transferred to the glassy carbon electrode or Nafion membrane?*

[Response 4] We agree with the reviewer's concern regarding the potential damage to the catalyst during the etching process of the Cu foil and the accuracy of the amount of Ir loading transferred to the target substrates (glassy carbon electrode or Nafion membrane). First of all, the total amount of Ir loading in each catalyst was obtained through inductively coupled plasma mass spectrometer (ICP-MS) experiments, which were conducted at least five times by using catalysts formed on Cu foil sacrificial substrates, not the target substrates. Therefore, further experiments were conducted to prove that there is no damage to the three-dimensional nanostructures and no change in the amount of Ir loading during the etching and transferring processes. We carried out additional SEM and ICP-MS analyses for both Ir/D-ATO fabricated on Cu foil and Ir/D-ATO transferred to Nafion membrane through the Cu foil etching (**Supplementary Fig. 13a**). As demonstrated in **Supplementary Fig. 13b** and **Fig. 13c**, it can be confirmed that the overall morphology and structure of Ir/D-ATO are well preserved during the etching of Cu foil and transferring to Nafion membrane. (Please note that the shape of the nanostructures in the Nafion membrane sample appears slightly thicker than that in the Cu foil sample because a thin Pt coating was applied to prevent the charging and damage of Nafion membrane during SEM analysis.) These SEM images provides direct evidence that the etching process of Cu foil does not damage the nanostructured catalysts.

The amount of Ir loading of transferred Ir/D-ATO on Nafion membrane was analyzed through ICP-MS and the results were plotted in **Supplementary Fig. 13d**. This result confirmed that the transferred Ir/D-ATO on Nafion membrane has an average loading amount of Ir value of $2.08 \mu\text{g cm}^{-2}$. This value is almost identical to the value of Ir/D-ATO formed on Cu foil ($2.023 \mu\text{g cm}^{-2}$), which was measured and reported in our original manuscript, previously. Therefore, ICP-MS results also provide direct evidence to prove that there is no change in the amount of Ir loading during the etching and transferring processes. Consequently, the amount of Ir loading on the Cu foil can represent the amount of Ir loading on the target substrates. In response to the reviewer's comment, we updated our manuscript to provide more details information about the absence of structural damage and variation of Ir loading amount after etching and transferring.

[Modification in the manuscript]

(Result, page 8) “The amount of Ir loading in Ir/D-ATO was determined through inductively coupled plasma mass spectrometry (ICP-MS) using samples formed on Cu foil. The Ir loading amount on Cu foil can accurately represent the amount of Ir loading on the glassy carbon electrode or Nafion membrane. This is due to the absence of damage during the etching of Cu foil and the subsequent transfer processes (Supplementary Fig. 13). The mass activity ~”

(Supplementary Information)

Supplementary Fig. 13. Characteristics of Ir/D-ATO before and after etching and transfer process from Cu foil to Nafion membrane. **a** Photographs of Ir/D-ATO fabricated on Cu foil (upper) and Nafion membrane (lower). Scanning electron microscopy (SEM) images of **b** Ir/D-ATO fabricated on Cu foil and **c** Nafion membrane with 2 μm scale bar. **d** The amount of Ir loading of Ir/D-ATO fabricated on Cu foil and Nafion membrane. The reported loading amount of Ir on the Nafion membrane represents the mean \pm standard deviation ($n = 3$).

[Comment 5] What caused the ECSA error for Ir/CNW to be significantly larger than for other catalysts in Fig. 3c?

[Response 5] We appreciate the reviewer's important comment regarding the error in the electrochemically active surface areas (ECSA) for Ir/CNW. From an ECSA perspective, the yield of the solvent-assisted nanotransfer printing (S-nTP) process, which we employed to fabricate three-dimensional nanostructures, is considerably significant. Furthermore, the yield of the S-nTP process is intimately associated with the surface energy of the target substrates. In particular, the yield of the S-nTP process tends to be higher when the surface energy of the target substrate is higher^{1,2}. Typically, metal oxides possess hydrophilic surface properties and correspondingly high surface energy (1.72 J m⁻² for SnO₂)³. However, the graphite carbon utilized as a support material in Ir/CNW displays hydrophobic properties, and it carries a relatively low surface energy of 0.228 J m⁻²⁴. When the same Cu foil serves as the target substrate for stacking the Ir/D-ATO building blocks, the increase in substrate's surface energy after the initial transfer ensures the effective transfer of subsequent layers. On the contrary, stacking Ir/CNW reduces the substrate's surface energy due to the hydrophobic properties of graphite carbon, resulting in a lower transfer yield for subsequent layers. This, in turn, causes a larger standard deviation in ECSA for Ir/CNW compared to other catalysts based on metal oxide supports.

References

1. Jeong JW, *et al.* High-resolution nanotransfer printing applicable to diverse surfaces via interface-targeted adhesion switching. *Nat. Commun.* **5**, 5387 (2014).
2. Nam TW, *et al.* Thermodynamic-driven polychromatic quantum dot patterning for light-emitting diodes beyond eye-limiting resolution. *Nat. Commun.* **11**, 3040 (2020).
3. Wang HW, *et al.* Structure and Stability of SnO₂ Nanocrystals and Surface-Bound Water Species. *J. Am. Chem. Soc.* **135**, 6885-6895 (2013).
4. Han Y, Lai KC, Lii-Rosales A, Tringides MC, Evans JW, Thiela PA. Surface energies, adhesion energies, and exfoliation energies relevant to copper-graphene and copper-graphite systems. *Surf. Sci.* **685**, 48-58 (2019).

[Comment 6] The authors claimed that the increase of O_2^- anion concentration caused a decrease of Ir charge states (the peaks moving to the left in Fig. 4e). Why did the O charge states also decrease simultaneously?

[Response 6] We thank the reviewer for raising this comment. In **Fig. 4c**, we showed that the increase of O_2^- concentration in EER indeed boosts the electron transfer from the ATO slab to the Ir_4O_8 cluster. The donated electrons from the support (EER-contained ATO slabs) would alter the charge states of both Ir and O atoms in the Ir_4O_8 cluster. As a result, these donated electrons decrease the charge states of both Ir and O species. Please note that, the reduced Ir charge state does not originate from the change of O atom within Ir_4O_8 cluster, but results from the donated electrons from the external system (EER).

Fig. 4. Charge transfer interaction between the Ir catalyst and ATO support with varying amounts of EER. **a** Characterization of $\sim c$ O_2^- anion-induced electron transfer plotted along the c -axis, or $\Delta\rho = \rho_{\text{total}} - [\rho_{\text{EER-contained support}} + \rho_{\text{catalyst}}]$. **d** 3D visualizations of the charge transfer for the four model systems. **e** Population densities for atoms in the Ir_4O_8 cluster with varying O_2^- concentrations. The cases termed ‘without support’ (the first row) refer solely to the Ir_4O_8 cluster.

[Modifications in the manuscript]

(Result, page 11) “EER donates electrons to the Ir_4O_8 cluster, which results in decreases in the charge states of both Ir and O species. The increase of \sim ”

[**Comment 7**] *There are some expression errors in this manuscript, such as the fraction of each component of the Ir/D-ATO in page 7. The authors should carefully check the full text and correct them. Besides, please make consistency with the nomenclature, specifically Ir black or IrB, in all the figures.*

[**Response 7**] We are grateful to the reviewer for pointing out these errors and issues. We meticulously reviewed and corrected errors in the manuscript.

[**Modifications in the manuscript**]

- The nomenclature (Ir black) of Fig. 3, 5 and Supplementary Fig. 9, 12, 14, 15, and 17 were updated.

(**Abstract, page 2**) “Suppressing the oxidation of active-Ir(III) in IrO_x catalysts is highly desirable to realize an efficient and durable oxygen evolution reaction (OER) in water electrolysis.”

(**Abstract, page 2**) “Both computational and experimental analyses reveal that the promoted charge transfer driven by EER is the key parameter for stabilizing the active-Ir(III) in IrO_x catalysts.”

(**Result, page 8**) “We also measured the OER activity of the commercial catalysts (Ir/C, Ir black) and Ir catalysts supported on a graphite carbon nanostructure (Ir/CNW).”

(**Method, page 16**) “All of the potentials indicated in electrochemical data were converted to the RHE scale ($E_{\text{RHE}} = E_{\text{Ag/AgCl}} + 0.059 \cdot \text{pH} + E^{\circ}_{\text{Ag/AgCl}}$, where $E^{\circ}_{\text{Ag/AgCl}}$ is 0.197 V at room temperature) and 100% iR-correction was conducted to compensate for ohmic drop in the electrolyte.”

(**Method, page 17**) “Electrochemical impedance spectroscopy (EIS) was conducted at 1.55 V_{RHE} from 1 Hz to 100 kHz. The intercept with x-axis at a high frequency region in the Nyquist plot (34 – 37 Ω) was used as electrolyte resistance for iR-correction.”

(**Fig. 4. Caption, page 22**) “**Fig. 4. Charge transfer interaction between the Ir catalyst and ATO support with varying amounts of EER. a** Characterization of ~”

(**Fig. 5. Caption, page 23**) “**Fig. 5. Electrochemical measurements of Ir/D-ATO, Ir/TiO₂, and Ir black in PEMWE cell. a** Polarization curves ~, **b** mass activity comparison estimated at 1.6 and 1.8 V (L-: low loading, 10 μg_{Ir} cm⁻², H-: high loading, 500 μg_{Ir} cm⁻²), **c** electrochemical ~”

Reviewer #2

[General review] *Response to authors: Accept with Major revisions*

The manuscript by Lee and co-workers explores the preparation of supported IrO_x catalyst onto an antimony-doped tin oxide with an excess of surface superoxide moieties to stabilize the Ir³⁺ active phase under oxygen evolution reaction conditions. Besides physicochemical characterization (STEM, XPS, XRD), the authors performed long-term durability testing on a PEMWE cell yielding promising results in terms of stability. Despite the aforementioned, there are several scientific questions to be addressed before consideration for publishing in Nature Communications:

[General response] We sincerely appreciate the reviewer's positive evaluation and attention paid to our manuscript, which has significantly helped to improve it. We have made attempts to fully address the reviewer's comments in the revised manuscript, and our point-by-point responses are provided below.

[Comment 1] *Despite the thorough XPS spectra deconvolution, the authors have disregarded the well-known asymmetric nature of the Ir 4f spectra. This, firstly postulated by Doniach and Šunjić (J. Phys. C.: Solid State Phys., 1970, 3, 285-291), has been adopted by the XPS community by employing Functional Lorentzian (LF) lineshapes. This can result in an underestimation of the main peak contribution (Ir³⁺ and Ir⁴⁺) to the overall spectral signal. In addition, the contribution of the Ir 5p ½ peak should be included in the peak deconvolution, generally found in the 64-65 eV range. Such contribution will unambiguously affect the Ir³⁺/Ir⁴⁺ analysis, as the Ir³⁺ doublet lies in such window. The authors should correct this by using the peak deconvolution lineshapes suggested by Freakley et al. (Surf. Interface Anal. 2017, 49, 794–799) and Pfeifer et al. (Surf. Interface Anal. 2016, 48, 261-273) to interpret the data. Authors should cite the following relevant works of such topic: Energy Environ. Sci., 2013, 6, 3756–3764; Phys. Chem. Chem. Phys., 2016, 18, 2292; ACS Catal. 2021, 11, 15, 9300–9316.*

[Response 1] We greatly appreciate the reviewer's insightful comment concerning the deconvolution of Ir 4f spectra. We acknowledge, as the reviewer indicated, the importance of the well-known asymmetric nature of Ir 4f spectra and the contribution of the Ir 5p ½ peak. In consideration of these two factors, we have deconvoluted all Ir 4f spectra in our manuscript afresh, thereby effectively excluding the possibility

of underestimating the contribution of Ir(III) and Ir(IV) peaks. Based on the peak deconvolution standards suggested in the two references, we adopted the Functional Lorentzian (LF) lineshape, and the intensity of the Ir 5p $\frac{1}{2}$ peak was set to be less than 5% of the total intensity^{1,2}. As a result, the fully deconvoluted XPS spectra were depicted in **Fig. 2f**, **Supplementary Fig. 17** (Ir/C and Ir black), and **Supplementary Fig. 18** (Ir/D-ATO, Ir/M-ATO, and Ir/S-ATO). Furthermore, the characterization of the oxidation states of Ir in Ir/D-ATO, Ir/M-ATO, and Ir/S-ATO at each sample (as-synthesized, post-activation, and post-ADT) plotted in **Fig. 4a** was also updated based on the newly analyzed Ir 4f spectra. However, despite this new processing, there was no change of data trend that the proportion of Ir(III) species consistently increased with EER content after activation and ADT, and Ir/D-ATO exhibited the highest Ir(III) portion of 20.69 % (post-activation) and 28.53% (post-ADT). Once again, we appreciate the reviewer's valuable comment, which guided us to analyze the XPS data more precisely. The manuscript and Supplementary Information were revised and updated to reflect the reviewer's comments.

References

1. Freakley SJ, Ruiz-Esquius J, Morgan DJ. The X-ray photoelectron spectra of Ir, IrO₂ and IrCl₃ revisited. *Surf. Interface Anal.* **49**, 794-799 (2017).
2. Pfeifer V, *et al.* The electronic structure of iridium and its oxides. *Surf. Interface Anal.* **48**, 261-273 (2016).

[Modifications in the manuscript]

(Result, page 7) “By deconvoluting the Ir 4f peaks into metallic Ir, Ir(III), and Ir(IV), while considering factors such as the Functional Lorentzian (LF) lineshape and Ir 5p $\frac{1}{2}$ peak, it was confirmed that the fractions of each component were 55.10%, 4.92%, and 39.28%, respectively^{43,44}. Most of ~”

(Result, page 10 – 11) “Furthermore, after the electrochemical activation, the ratio of Ir(III) species increased with the EER content, and Ir/D-ATO exhibited the highest Ir(III) portion (20.69%). In particular, the Ir(III) to Ir(IV) ratio ($R_{III/IV}$) of Ir/D-ATO was calculated to be 0.563, which corresponds to approximately 1.44 times and 1.77 times that of Ir/M-ATO and Ir/S-ATO, respectively.”

(Result, page 11) “For the case of Ir/D-ATO, not only did the overall portion of Ir(III) increase, but the ratio of Ir(III) to Ir(IV) also significantly rose to 0.757, which is 1.34 times higher than the value of the post-activation sample. On the other hand, ~”

Fig. 2. Fabrication and characterization of dense EER ATO supported Ir catalyst (Ir/D-ATO). a Schematic illustration ~. e XRD spectra of Ir/D-ATO before and after high temperature annealing. f XPS spectra of the Ir 4f level on Ir/D-ATO.

Fig. 4. Charge transfer interaction between the Ir catalyst and ATO support with varying amounts of EER. a Characterization of the oxidation states of Ir in Ir/D-ATO, Ir/M-ATO, and Ir/S-ATO for each sample (as-synthesized, post-activation, post-ADT) by XPS. ~

(Supplementary Information)

Supplementary Fig. 17. X-ray photoelectron spectroscopy (XPS) spectra of the Ir 4f level on (a, b) Ir/C, and (c, d) Ir black attained in the post-activation and post-ADT samples to compare the ratio of metallic Ir, Ir(III), and Ir(IV) of the catalysts.

Supplementary Fig. 18. X-ray photoelectron spectroscopy (XPS) spectra of the Ir 4f level on (a-c) Ir/D-ATO, (d-f) Ir/M-ATO, and (g-i) Ir/S-ATO attained in the as-synthesized, post-activation, and post-ADT samples to compare the ratio of metallic Ir, Ir(III), and Ir(IV) of the catalysts.

[Comment 2] *Following up with the XPS characterization, the authors ascribe a Sn satellite at higher binding energies upon increasing superoxide contents on ATO. Could the authors provide further insights on this? I would have anticipated, unless a total formal charge is present in ATO, that either Sb or Sn are present in a lower oxidation state. However, previous XPS studies would suggest the aforementioned satellite would correspond indeed to Sn⁴⁺ and the Sn⁴⁺ peak to that of Sn²⁺ (Kwoka et al., Thin Solid Films 2005, 490, 36-42) although it is hard to tell given that there is no information on the exact deconvoluted peak binding energies, which should be provided in the revised version of the manuscript. In addition to this, can the authors explain why are there no differences in Ir chemical states in pristine samples with varied EER contents?*

[Response 2] We are grateful to the reviewer for raising this comment regarding the Sn satellite at higher binding energies. First of all, as the reviewer rightly observed, the Sn satellite peak is mistakenly located at Sn⁴⁺ and the Sn⁴⁺ peak at Sn²⁺ in our original XPS characterization results. This error resulted from a miscalibration during the peak calibration process using the carbon peak. We failed to notice this calibration error and acknowledge our mistakes in the XPS data. However, in our previous study, we correctly reported the shift in the Sn 3d peak's binding energy due to oxygen adsorption on the tin oxide surface¹.

To address the concern raised by the reviewer, ATO nanostructures were newly prepared and additional XPS analysis was performed. As a result, we obtained the same correctly positioned Sn 3d peak as described in the aforementioned paper. (**Supplementary Fig. 5**). Compared to the bulk sample, where only the 486.6 eV peak appears, the Sn 3d peaks of EER-contained ATO samples can be deconvoluted into Sn^{IV} and Sn^{Satellite} peaks, which are assigned to 486.6 eV and higher binding energy than 486.6 eV, respectively. In response to the reviewer's comment, these results were updated in the revised manuscript and Supplementary Information to correct the mistakes related to the peak binding energies.

The reviewer also requested further insights on this XPS data. Therefore, in order to understand its electronic origin, we performed DFT calculations and Bader charge analysis for the systems with and without EER, as shown in **Fig. R2**. Our analysis revealed that Sn atoms in the ATO slab with EER lose more electrons (on average 0.018 e⁻/Sn atom) compared to those of the bare ATO slab, which well agrees with experimental observations that Sn satellite peaks become stronger for more superoxide (O₂⁻) contents in EER. (**Supplementary Fig. 5**).

The additional question regarding the Ir chemical states in pristine samples with varied EER contents can be responded clearly through the points discussed in [Response 1]. From the corrected XPS spectra in **Supplementary Fig. 18** and the oxidation states summarized in **Fig. 4a**, there is a slight difference in the ratio among metallic Ir, Ir(III), and Ir(IV) of the as-synthesized samples. Specifically, Ir/D-ATO exhibited the highest Ir(III) portion of 4.92%, which corresponds to approximately 1.17 times and 1.22 times those of Ir/M-ATO and Ir/S-ATO, respectively. As a result of reflecting the asymmetric nature of Ir 4f spectra and the contribution of the Ir 5p $\frac{1}{2}$ peak to the revised peak deconvolution method, the difference of Ir (III) portion in the as-synthesized sample was clearly identified and these results are added and updated in the revised version of manuscript and Supplementary Information to address the reviewer's comment.

Fig. R2. Comparison of charge states of species in ATO slabs with and without EER, based on Bader charge analysis. The top view images of bare ATO (without EER) and EER-ATO (with EER) are shown on top. The table in the bottom summarizes the charge states of each Sn, Sb, and O atoms (within ATO slab) for both bare ATO and EER-ATO.

Reference

1. Han HJ, *et al.* Unconventional grain growth suppression in oxygen-rich metal oxide nanoribbons. *Sci. Adv.* **7**, eabh2012 (2021).

[Modifications in the manuscript]

(Result, page 10) “Moreover, as depicted in Fig. 4a, there is a small variation in the proportion of Ir(III) in the as-synthesized samples, and a relatively higher Ir(III) portion is exhibited for Ir/D-ATO. Furthermore, after the electrochemical activation, ~”

Fig. 4. Charge transfer interaction between the Ir catalyst and ATO support with varying amounts of EER. a Characterization of the oxidation states of Ir in Ir/D-ATO, Ir/M-ATO, and Ir/S-ATO for each sample (as-synthesized, post-activation, post-ADT) by XPS. ~

(Supplementary Information)

Supplementary Fig. 5. X-ray photoelectron spectroscopy (XPS) spectra of Bulk and EER contained ATO nanowire arrays with varying density of EER. The binding energies of Sn 3d spectra (a, c, e, g) and O 1s spectra (b, d, f, h) of each sample, respectively.

[Comment 3] *Can the authors report the XPS spectra of Ir/CNW? This would be the easiest approach to prove the CSI effect of ATO versus an alternative support with the same Ir synthetic method.*

[Response 3] We agree with the reviewer's comment that reporting the XPS spectra of Ir/CNW is a good way to prove the strong-metal support interaction of EER-contained ATO. To compare the oxidation states of Ir with Ir/D-ATO, we performed additional XPS measurements using Ir/CNW and obtained Ir 4f spectra of Ir/CNW in each sample (as-synthesized, after-activation, and after-ADT). After that, acquired spectra were deconvoluted for analyzing the ratio of metallic Ir, Ir(III), and Ir(IV). **Supplementary Fig. 19** plotted the deconvoluted XPS spectra of Ir/D-ATO and Ir/CNW for the comparison and the characterization of the oxidation states of Ir in Ir/D-ATO and Ir/CNW was summarized in **Supplementary Fig. 20**. As expected, these two catalysts exhibited a large difference of the proportion of Ir(III) for the after-activation and after-ADT samples, resulting from the catalyst-support interaction. After the electrochemical activation, Ir/D-ATO showed the Ir(III) portion of 20.69%, whereas Ir/CNW showed only 11.53%. More specifically, $R_{III/IV}$ of Ir/D-ATO is 2.16 times higher than Ir/CNW. This result indicates that more electrons are supplied from the supports to the catalysts due to the catalyst-support interaction when EER-contained ATO is used as the support material. The difference in the oxidation states of Ir, especially Ir(III), between Ir/D-ATO and Ir/CNW becomes more pronounced in the after-ADT sample. Both the overall portion of Ir(III) and $R_{III/IV}$ was significantly higher in the case of Ir/D-ATO, while Ir/CNW exhibited a decrease of $R_{III/IV}$ after the ADT. In conclusion, despite the method of forming Ir catalysts is the same, it was confirmed that the CSI effect depends on the type of support materials. In response to the reviewer's comment, we added XPS spectra of Ir/CNW in Supplementary Information of our revised manuscript.

[Modifications in the manuscript]

(Result, page 10) "The Ir 4f spectra were deconvoluted to compare the ratio of metallic Ir, Ir(III), and Ir(IV) of the catalysts attained in each sample: as-synthesized, post-activation, and post-accelerated degradation test (ADT) with Ir/ATOs as well as the other catalysts (Supplementary Fig. 17, 18, and 19)⁴³. First, from the comparison of the oxidation states of Ir/D-ATO and Ir/CNW summarized in **Supplementary Fig. 20**, it was confirmed that, despite that the same method is employed for forming the Ir catalysts, the catalyst-support interaction (CSI) effect significantly depends on the type of support materials. More electrons are supplied from the EER-contained

ATO to the catalysts compared to graphite carbon, which leads to a high Ir(III) ratio across all stages of the samples. Moreover, as depicted in ~”

(Supplementary Information)

Supplementary Fig. 19. X-ray photoelectron spectroscopy (XPS) spectra of the Ir 4f level on (a-c) Ir/D-ATO, (d-f) Ir/CNW attained in the as-synthesized, post-activation, and post-ADT samples to compare the ratio of metallic Ir, Ir(III), and Ir(IV) of the catalysts.

Supplementary Fig. 20. Characterization of the charge transfer capability according to the support materials. Characterization of the oxidation states of Ir in **a** Ir/D-ATO and **b** Ir/CNW for each sample (as-synthesized, post-activation, and post-ADT) by XPS.

[Comment 4] *In order to report electrochemically active surface areas, the authors employ here CO stripping. The authors should justify why CO stripping was employed instead of more consolidated methods in the literature for Ir-based catalysts such as Hg UPD (Alia et al., Journal of The Electrochemical Society, 163 (11) F3051-F3056 (2016)), Zn cation adsorption (Zhao et al., Journal of The Electrochemical Society, 162 (12) F1292-F1298 (2015)) or adsorption capacitance measurements (Watzel et al., ACS Catal. 2019, 9, 9222–9230). CO stripping is based upon substrate-specific interactions, and there is no proof that CO adsorption selectively takes place only on top of Ir and not on Sb or Sn.*

[Response 4] We appreciate the reviewer’s comment regarding the method for estimating ECSA and agree with the concern regarding the CO adsorption selectivity on top of Ir and not on Sb or Sn. First of all, the CO stripping method is widely used to evaluate the electrochemically active surface area of Ir-based catalysts^{1,2,3} along with Hg UPD, Zn cation adsorption, or adsorption capacitance measurements method suggested by the reviewer. All of the reference papers suggested by the reviewer also mention the ECSA measurement through CO stripping.

The reason we chose CO stripping as the ECSA measurement method in this study is due to its relatively larger specific charge (Coulombic charge) value compared to Hg or H₂. The current peak that emerges in a specific potential range in the cyclic voltammetry curve originates from the adsorption/desorption of a

monolayer of certain molecules on the catalyst surface. As a result, the Coulombic charge value of these molecules plays a dominant role in determining the intensity of the resulting current peak. The Coulombic charge of the CO molecule is $420 \mu\text{C cm}^{-2}$, yielding a higher intensity of current peak than Hg ($138.6 \mu\text{C cm}^{-2}$) or H_2 ($179 \mu\text{C cm}^{-2}$)^{4,5}. The higher the intensity of the current peak, the easier it is to measure the ECSA of catalysts with an extremely low Ir loading as in our samples, and the measurement error also can also be minimized. Another reason for using CO stripping is that the catalyst is formed by e-beam deposition of Ir metal rather than Ir oxide. Zn cation adsorption is an alternative method to overcome the limitation of other methods that cannot be applied to IrO_2 ⁶. However, it is a less widely used method for Ir metal and was thus not chosen for our study. Lastly, we did not use adsorption capacitance measurements because they could be affected not only by Ir but also Sn or Sb used in the support material³.

In response to the reviewer's comment, we prepared two samples, one Ir/D-ATO, and another D-ATO, and conducted further comparative experiments. The cyclic voltammetry curves of Ir/D-ATO and D-ATO after the CO stripping process are described in **Fig. R3**. The current peak arising from the 0.8 to 1.15 V_{RHE} range corresponds to the CO stripping charge. A clear CO stripping peak was observed in the Ir/D-ATO, but the peak intensity of D-ATO was negligible. Based on these results, it is evident that the CO stripping peak is minimally influenced by Sn or Sb present in the support. Consequently, we have concluded that the CO stripping method is a valid approach for measuring the ECSA of Ir-based catalysts including Ir/D-ATO.

Fig. R3. CO stripping voltammetry curves and cyclic voltammetry curves of **a** Ir/D-ATO, **b** D-ATO to calculate the ECSA estimated in 0.05 M H_2SO_4 .

References

1. Kwon T, *et al.* Cobalt Assisted Synthesis of IrCu Hollow Octahedral Nanocages as Highly Active Electrocatalysts toward Oxygen Evolution Reaction. *Adv. Funct. Mater.* **27**, 1604688 (2017).
2. Park J, Sa YJ, Baik H, Kwon T, Joo SH, Lee K. Iridium-Based Multimetallic Nanoframe@Nanoframe Structure: An Efficient and Robust Electrocatalyst toward Oxygen Evolution Reaction. *ACS Nano* **11**, 5500-5509 (2017).
3. Watzele S, *et al.* Determination of Electroactive Surface Area of Ni-, Co-, Fe-, and Ir-Based Oxide Electrocatalysts. *ACS Catal.* **9**, 9222-9230 (2019).
4. Alia SM, Hurst KE, Kocha SS, Pivovar BS. Mercury Underpotential Deposition to Determine Iridium and Iridium Oxide Electrochemical Surface Areas. *J. Electrochem. Soc.* **163**, F3051-F3056 (2016).
5. Durst J, Simon C, Hasche F, Gasteiger HA. Hydrogen Oxidation and Evolution Reaction Kinetics on Carbon Supported Pt, Ir, Rh, and Pd Electrocatalysts in Acidic Media. *J. Electrochem. Soc.* **162**, F190-F203 (2015).
6. Zhao S, *et al.* Calculating the Electrochemically Active Surface Area of Iridium Oxide in Operating Proton Exchange Membrane Electrolyzers. *J. Electrochem. Soc.* **162**, F1292-F1298 (2015).

[Comment 5] *Regarding the authors statement on Page 8 ‘In general, it is well known that the interaction between the carbon support and catalysts is too weak to augment the intrinsic activity of catalysts¹⁷’, it is well known in the OER community that carbon is not an adequate support for acidic oxygen evolution electrocatalysis as it undergoes corrosion at anodic potentials (Yi *et al.*, *Catalysis Today* 2017, 295, 32-40; Geiger *et al.*, *ChemSusChem* 2017, 10, 4140-4143; Browne *et al.* *J. Mater. Chem. A* 2018, 6, 14162-14169; Edginton *et al.*, *ACS Appl. Energy Mater.* 2022, 5, 10, 12206–12218). Thus, the authors should take this into account when evaluating the electrochemical data, particularly relevant in the poor stability of Ir/C (Supplementary Fig. 13), which surely stems from carbon corrosion and subsequent Ir particle detachment from the working electrode.*

[Response 5] We appreciate the reviewer’s careful comment. As the reviewer pointed out, we agree that carbon is not a suitable material for support in the field of oxygen evolution electrocatalysis due to the poor stability driven by carbon corrosion and subsequent Ir particle detachment from the working electrode. Actually, glassy carbon electrodes cannot be used for stability tests under harsh conditions such as 10 mA cm⁻² of chronopotentiometry measurement. This is because when a potential higher than a specific value (> 2 V) is applied, degradation such as cracking and burning on the glassy carbon electrode occurs. Therefore, to overcome these issues, we used a Ti plate as a substrate of the working electrode at 10 mA cm⁻² condition in this study. We did not include the substrate information in our original manuscript, and in response to the reviewer's comments, we have added these details, along with additional references and sentences, in the revised manuscript.

[Modifications in the manuscript]

(Result, page 9) “It is well known that the glassy carbon working electrode can also suffer damage at high current density, which affects the stability of catalysts^{47,48}. To mitigate carbon corrosion and detachment of Ir catalysts from the backing electrode under harsh conditions, a Ti plate was used as a substrate. When such a high current density was applied to the catalysts ~”

(Method, page 17) “~ with a rotation speed of 1600 rpm. At 1 and 10 mA cm⁻² conditions, a glassy carbon working electrode and Ti plate were used as substrates, respectively. In addition, ~”

[Comment 6] *Electrocatalyst assessment based purely on potentiostatic holds are well-known to provide misleading metrics in terms of electrocatalyst stability as catalyst layer quality, active site blockage by microbubbles or surface oxidation state changes ultimately affect the experimental results. The following works discuss these issues at length, which should be discussed and cited in the manuscript: Kibsgaard et al., Nat. Energy 2019, 4, 430-433; R. Frydendal et al. ChemElectroChem 2014, 1, 2075; Lazaridis et al., Nat. Catal. 2022, 5, 363-373; Ehelebe et al., Curr. Opin. Electrochem. 2021, 29, 100832. The authors should evaluate the catalyst stability by quantifying dissolved amounts of Ir/Sb/Sn in the acidic electrolyte with ICP-MS to unambiguously prove that indeed the catalyst-support interactions stabilize the Ir³⁺ active sites.*

[Response 6] We acknowledge the reviewer's insightful comment on the evaluation of the electrochemical stability of Ir/D-ATO. As the reviewer indicated, various factors such as catalyst layer quality, active site blockage by microbubbles, or surface oxidation state changes need to be considered when assessing the durability of catalysts. Through the SEM image to be shown in **[Response 8]**, it can be seen that the highly ordered three-dimensional nanoarchitectures of Ir/D-ATO are uniformly formed on the entire target substrate. These three-dimensional nanoarchitectures can give long-range connectivity inside the catalyst layers for high electronic conductivity. Thus, the catalyst layer quality of the Ir/D-ATO is much better than that of conventional nanoparticle-type catalysts such as Ir/C or Ir black. Moreover, since the open structure with well-defined pores of Ir/D-ATO highly promotes the mass transport of reactants or generated gas bubbles¹ (our previous paper), the active site blockage issue would be minimized in our cross-stacked 3D nanostructures.

As demonstrated in this study, oxidation state modulation of Ir catalysts is the key parameter in maximizing the electrochemical stability of Ir/D-ATO, but it is driven by the charge replenishment through interaction with the metal oxide support. Due to the nature of the oxygen evolution reaction, oxidation reactions mainly occur. However, maintaining a low oxidation state of Ir by charge replenishment even in this environment means that the changes of oxidation state caused by the applied potential during the stability test have a negligible effect on stability. Various studies, including references presented by the reviewer, already pointed out the reliability issue of the half-cell-based stability measurement because of these problems, and accordingly, and therefore stability tests using single-cell devices are more widely accepted. In the revised manuscript, we provide additional stability results using a single-cell PEMWE system operating in more harsh conditions. The detailed information is provided in **Reviewer #1 – [Response 2]**.

Nevertheless, we agree with reviewer's concerns, and in response, we performed additional half-cell stability experiments using ICP-MS to quantify the dissolved amounts of Ir/Sb/Sn in the electrolyte after the stability measurement. After 15 hours of the chronopotentiometry test, we collected a portion of the electrolyte to calculate the dissolved amounts of elements in Ir/D-ATO catalyst. As a result, the total amounts of dissolved Ir, Sn, and Sb during the stability test were measured to be 0.087, 0.176, and 0.575 μg , respectively. Especially, Ir/D-ATO exhibited a significantly lower Ir mass loss of about 21.3% of the initial Ir loading compared to the commercial catalysts such as IrO_x/C known to show an Ir mass loss of at least 60% under the same stability test conditions². This indicates that Ir/D-ATO possesses high corrosion resistance toward Ir dissolution during OER despite its large ECSA of $132 \text{ m}^2 \text{ g}^{-1}$. Consequently, we believe that the charge transfer driven by EER leads to superior durability of Ir/D-ATO by preventing the oxidation of active-Ir(III), which can effectively suppress the dissolution of Ir.

References

1. Kim YJ, *et al.* Highly efficient oxygen evolution reaction via facile bubble transport realized by three-dimensionally stack-printed catalysts. *Nat. Commun.* **11**, 4921 (2020).
2. Oh HS, *et al.* Electrochemical Catalyst-Support Effects and Their Stabilizing Role for IrO_x Nanoparticle Catalysts during the Oxygen Evolution Reaction. *J. Am. Chem. Soc.* **138**, 12552-12563 (2016).

[Modifications in the manuscript]

(Result, page 9) “Mostly, degradation of catalysts is caused by dissolution and detachment during the oxidation of the Ir surface^{15,49,50}. To quantify the absolute Ir dissolution after the stability measurement, we measured the potential for 15 hr at 1 mA cm⁻². The dissolved Ir in the electrolyte was evaluated using ICP-MS, which confirmed a low Ir mass loss of about 21.3% of the initial Ir loading. Charge transfer of the EER-contained support can prevent the oxidation of Ir catalysts and serves to maintain them, which can effectively suppress the dissolution of Ir, leading to superior durability of Ir/D-ATO.”

[Comment 7] XPS characterization of Sb and Sn should be provided after ADT testing to showcase if there are any significant oxidation state changes at the support as well as any presence of the superoxides obtained during oxygen-rich evaporative deposition conditions

[Response 7] We agree with the reviewer's comment that XPS spectra of Sn and Sb after ADT testing should be provided to verify the oxidation state changes at the support. In response to the reviewer's comment, we conducted further experiments to obtain Sn 3d and Sb 3d spectra using Ir/D-ATO before (as-synthesized) and after ADT testing. First of all, in the case of the Sb 3d peak, no change such as the peak position before and after the ADT test was verified (**Supplementary Fig. 21b, d**), which means that the oxidation state of Sb does not change during the ADT test. On the contrary, as shown in **Supplementary Fig. 21a and 21c**, a slight peak shift and a change in the ratio of deconvoluted peaks can be confirmed in the Sn 3d spectra. The Sn satellite peak located at higher binding energy than 486.6 eV originates from the EER at the surface of ATO, which is already discussed in **[Response 2]**. Before the ADT test, the Sn 3d peak of Ir/D-ATO exhibited a large portion of Sn^{Satellite} compared to Sn^{IV}. However, after the ADT, it can be seen that the overall peak shifts to the lower binding energy as the intensity of the Sn^{Satellite} peak diminishes and the intensity of Sn^{IV} intensifies. The shape of the deconvoluted peak of Ir/D-ATO became similar to that of the Sn 3d peak of S-ATO as reported in **[Response 2]**, indicating that the density of EER decreased during ADT testing. Nevertheless, the superoxides obtained during e-beam deposition still exist because the Ir/D-ATO after ADT is located at higher binding energy and the Sn satellite peak is also more identifiable compared to the bulk sample in **Supplementary Fig. 5**. Accordingly, in response to the reviewer's comment, Supplementary Information and corresponding sentences were added to the revised manuscript.

[Modifications in the manuscript]

(Result, page 11) “Finally, Sn 3d and Sb 3d spectra were also analyzed to investigate any changes in the chemical composition of the support after the ADT, and it was verified that there was no noticeable difference, except for a slight shift (0.65 eV) of the Sn 3d peak compared to the initial status (**Supplementary Fig. 21**).”

(Supplementary Information)

Supplementary Fig. 21. X-ray photoelectron spectroscopy (XPS) spectra of the Sn 3d and Sb 3d of Ir/D-ATO (a, b) before and (c, d) after the ADT to confirm the oxidation state changes at the support.

[Comment 8] Single-cell performance in PEMWE: catalyst layer imaging for Ir/D-ATO should be provided to evaluate the quality of the spraycoating. In addition, the authors should have employed a commercial supported Ir catalyst such as Elyst75 (<https://fcs.unicore.com/en/fuel-cells/products/elyst-ir75-0480>) to provide a more realistic benchmarking instead of an unsupported Ir catalyst which clearly yielded inhomogenous catalyst layers at low loadings. Regarding the effect of Ir loadings in PEMWE performance, the authors should cite the work of Gasteiger's group (Bernt et al., *J. Electrochem. Soc.* 2018, 165, 5, F305-F314).

[Response 8] We are grateful for the reviewer’s constructive advice to appropriately emphasize our results. Before proceeding, we would like to clarify that our Ir/D-ATO samples were fabricated on the Nafion membrane using the transfer printing method, not the spray coating commonly used in other nanoparticle-type catalysts. As demonstrated in **Supplementary Fig. 20**, the quality of the Ir/D-ATO on the Nafion membrane was verified through SEM analysis. Three-dimensional nanostructures were uniformly fabricated across the entire area to produce homogeneous catalyst layers.

As the reviewer suggested, we purchased Elyst75, which is TiO₂-supported Ir catalyst (denoted as Ir/TiO₂ in the revised manuscript), and measured its performance in the PEMWE cell at Ir loadings of 10 and 500 μg_{Ir} cm⁻². The electrochemical measurements and material characterizations of Ir/TiO₂ are added in **Fig. 5** and Supplementary Information of the revised manuscript. Although it was supported on TiO₂, the Ir/TiO₂ was unable to form a homogeneous catalyst layer at 10 μg_{Ir} cm⁻², which is close to the Ir loading of Ir/D-ATO (7.2 μg_{Ir} cm⁻²), due to the much larger particle size, as shown in **Supplementary Fig. 21b and 23a**. Thus, its performance in the PEMWE cell was significantly lower than our Ir/D-ATO. The reason why we set the high Ir loading of 500 μg_{Ir} cm⁻² was actually based on the reference work suggested by the reviewer¹. The report mentioned that the catalyst layer becomes inhomogeneous at Ir loadings less than 500 μg_{Ir} cm⁻². In this regard, we provided the SEM images of the catalyst layers prepared at 500 μg_{Ir} cm⁻², confirming that the catalyst layers are sufficiently homogeneous (**Supplementary Fig. 23b**). We updated the figures / Supplementary Information and also cited the reference work in our revised manuscript.

Reference

1. Bernt M, Siebel A, Gasteiger HA. Analysis of Voltage Losses in PEM Water Electrolyzers with Low Platinum Group Metal Loadings. *J. Electrochem. Soc.* **165**, F305-F314 (2018).

[Modifications in the manuscript]

(Result, page 12) “Furthermore, we demonstrated the superior catalytic performance of Ir/D-ATO in a PEMWE cell through comparison with the commercial Ir black catalyst and TiO₂-supported Ir catalyst (Ir/TiO₂) (**Supplementary Fig. 24**). The anode catalyst layers with the commercial catalysts were prepared with two Ir loadings, one with similar Ir loading of Ir/D-ATO (10 μg_{Ir} cm⁻²

²) and another with a much higher Ir loading ($500 \mu\text{g}_{\text{Ir}} \text{cm}^{-2}$) (**Supplementary Fig. 27 and 28**). The higher Ir loading value was set to verify the formation of continuous catalyst layers.⁵¹ Despite the ultralow Ir loading ($7.2 \mu\text{g}_{\text{Ir}} \text{cm}^{-2}$) of Ir/D-ATO, the catalytic performance of Ir/D-ATO far exceeded those of the Ir/TiO₂ and Ir black samples (**Fig. 5a**), and the mass activity of Ir/D-ATO was 55 and 75 times higher than those of high-loaded Ir/TiO₂ and Ir black at 1.6 V, respectively (**Fig. 5b**).”

(Result, page 12) “In particular, at a low Ir loading, a uniformly connected 3D nanostructure of Ir/D-ATO (**Supplementary Fig. 25**) is desirable ~”

(Result, page 12) “As shown in **Supplementary Fig. 26**, the commercial Ir black and Ir/TiO₂ are composed of relatively larger nanoparticles (>100 nm), resulting in discontinuous and non-uniform catalyst layers at low Ir loadings (**Supplementary Fig. 27a and 28a**); it is highly likely ~”

(Result, page 12) “Consequently, the charge transfer resistance of the low-loaded Ir black and Ir/TiO₂ is considerably higher than that of the Ir/D-ATO (**Fig. 5c**), resulting ~”

(Result, page 12) “On the other hand, the high-loaded catalyst layers of the Ir black and Ir/TiO₂ are sufficiently continuous (**Supplementary Fig. 27b and 28b**) and demonstrate much higher catalytic performance than their corresponding low-loaded catalyst layers (**Supplementary Fig. 31a**). This is attributable to the reduced charge transfer resistance, as evidenced by the Nyquist plots of the highly-loaded catalyst layers, which are not notably different from that of Ir/D-ATO (**Supplementary Fig. 31b**).”

(Result, page 12) “Nevertheless, the Ir/D-ATO showed significantly higher mass activity than the high-loaded Ir black and Ir/TiO₂ samples; in addition to ~”

Fig. 5. Electrochemical measurements of Ir/D-ATO, Ir/TiO₂ and Ir black in PEMWE cell. a Polarization curves at low Ir loadings of $7.2 \mu\text{g}_{\text{Ir}} \text{ cm}^{-2}$ for Ir/D-ATO and $10 \mu\text{g}_{\text{Ir}} \text{ cm}^{-2}$ for Ir/TiO₂ and Ir black, **b** mass activity comparison estimated at 1.6 and 1.8 V (L:-low loading, $10 \mu\text{g}_{\text{Ir}} \text{ cm}^{-2}$, H:-high loading, $500 \mu\text{g}_{\text{Ir}} \text{ cm}^{-2}$), **c** electrochemical impedance spectroscopy data, **d** Ir-specific power vs. mass activity at 1.6 V plot with previous studies, and **e** chronopotentiometry data for stability test of Ir/D-ATO in PEMWE cell at 1.0 A cm^{-2} .

(Supplementary Information)

Supplementary Fig. 25. Scanning electron microscopy (SEM) image of Ir/D-ATO fabricated on Nafion membrane for single-cell measurement in PEMWE.

Supplementary Fig. 26. Transmission electron microscopy (TEM) images of **a** commercial Ir black and **b** Ir/TiO₂ catalyst.

Supplementary Fig. 28. Scanning electron microscopy (SEM) images of a low-loaded ($0.01 \text{ mg}_{\text{Ir}} \text{ cm}^{-2}$) and **b** high-loaded ($0.5 \text{ mg}_{\text{Ir}} \text{ cm}^{-2}$) Ir/TiO₂ catalyst layer.

Supplementary Fig. 31. Electrochemical measurements in PEMWE single cell. **a** Polarization curves and **b** electrochemical impedance spectroscopy data of high-loaded Ir/TiO₂ and Ir black.

[Comment 9 – Minor Point 1] *The authors fail to cite relevant references concerning the degradation mechanisms of Ir: Kasian et al., Angew. Chem. Int. Ed. 2018, 57, 2488-2491
Kasian et al., Energy Environ. Sci., 2019, 12, 3548-3555
Scott et al., Energy Environ. Sci., 2022,15, 1988-2001*

[Response 9 – Minor Point 1] We appreciate the reviewer’s kind feedback. We cited the relevant references in the revised manuscript.

[Modifications in the manuscript]

(Main, page 3) “During the oxidation of Ir(III), the formation of soluble intermediates leads to gradual Ir dissolution and degradation of the catalytic performance over time^{15,19,20}.”

[Comment 9 – Minor Point 2] *The statement on page 3, second paragraph ‘Alternatively, sustainably retaining active-Ir(III) species is considered one of the most desired strategies to overcome the fundamental trade-off relationship between activity and durability’ should be backed up by relevant references.*

[Response 9 – Minor Point 2] We appreciate the reviewer’s suggestion. We added the relevant references in the revised manuscript.

[Modifications in the manuscript]

(Main, page 3) “Alternatively, sustainably retaining active-Ir(III) species is considered one of the most desired strategies to overcome the fundamental trade-off relationship between activity and durability^{22,23}, ~”

[Comment 9 – Minor Point 3] *The section labelled as Discussion corresponds to the Conclusions of the manuscript. This should be amended in the revised version of the manuscript.*

[Response 9 – Minor Point 3] We understand the reviewer's concern about the formatting of section headings and have consulted the formatting instructions of Nature Communications to address this. Regrettably, the only permitted section headings in the main text are "Results" and "Discussion," excluding "Conclusions". We interpret the "Discussion" section as discussing our findings from the perspective of the overall manuscript. It has also been confirmed that recent articles in Nature Communications consistently adhere to this guideline. Therefore, we must keep the section headings in the current form to comply with the journal's guidelines.

Reviewer #3

[General review]

The authors present an anode catalyst for PEM water electrolysis consisting of IrO_x nanoparticles decorated on an ATO support. The ATO was grown by e-beam evaporation-deposition, and the content of O₂⁻ species could be controlled by variation in time, forming what the authors call a an excess electron reservoir (EER, O₂⁻). These supported catalysts were characterized and the conclusions are supported by DFT calculations and screening. The catalyst layer was then fabricated by solvent-assisted nanotransfer printing method allowing a high surface area 3D structure, which was tested in half-cell and single full cell. The catalyst showed remarkable Ir-specific power at 1.6 V using a very low Ir loading.

This work is interesting for the development of Ir catalysts supported on metal oxides and for the presented fabrication method of the anode catalyst layer for PEMWE. I recommend publications after revision.

[Response] We sincerely appreciate the reviewer's positive evaluation and attention paid to our manuscript. We try our best to address the reviewer's comments in the revised manuscript, and below are our point-by-point responses.

[Comment 1] *Abstract, line 26: "an excess electron reservoir (EER, O₂⁻)-decorated on antimony-doped tin oxide" When I first read the abstract it was not clear to me what the EER exactly is. I think this has to be clarified in the abstract. Also the use of "decorated on" is confusing, since it seems to me more as "incorporated in" as used also later in the text is more appropriate. My understanding is that this EER is just a part of the ATO film that presents a specific oxygen species (O₂⁻), and is created during the fabrication of the ATO layer, not exactly a surface decoration, which might confuse the reader implying that there is an heterostructure on top of the ATO.*

[Response 1] We appreciate the reviewer for this helpful suggestion to properly clarify the concept of the EER. We have included the details of the EER in the abstract for precise clarification. We also agree with this comment and have therefore replaced 'decorated on' with 'incorporated in' throughout the revised manuscript."

[Modifications in the manuscript]

(Abstract, page 2) “Here, we demonstrate that an excess electron reservoir (EER), which is a charged oxygen species, incorporated in antimony-doped tin oxide (ATO) supports can effectively control the Ir oxidation states by boosting the charge donations to IrO_x catalysts.”

(Abstract, page 2) “When used in a polymer electrolyte membrane water electrolyzer, Ir catalyst on EER-incorporated ATO support ~”

(Main, page 4) “The Ir catalysts placed on EER-incorporated ATO demonstrated significantly enhanced mass activity, ~”

[Comment 2] *Abstract, line 33: “remarkable potential for realizing gigawatt-scale H2 production” and page 12, line 337-338. I have a question regarding scalability, could the authors state in the text (not necessary in the abstract) what is the actual electrode size achievable with the nanotransfer technology and what are the promises of this interesting method.*

[Response 2] We are grateful to the reviewer for providing valuable comments regarding the scalability of nanotransfer printing technology. In this study, we demonstrate both half-cell measurements with RDE systems and single-cell measurements with PEMWE systems, with the actual electrode size of Ir/D-ATO catalysts being 1 cm². However, the nanotransfer printing technology that we used in this study can be applied to fabricate three-dimensional nanostructures on a wafer-scale. In our previous studies, we have already reported a method to increase scalability by fabricating nanowire arrays on a 6 – 8-inch wafer scale, as shown in **Fig. R4**^{1,2}. If an automated process such as roll-to-roll can be applied to the nanotransfer printing technology that has succeeded in wafer-scale, it will gain great attention as a promising catalysts fabrication method. In response to the reviewer’s comment, we added additional texts and reference in the revised manuscript.

Fig. R4. a Transfer-printed pattern on the 8-inch wafer on a transparent and flexible PET substrate. (*Sci. Adv.*, **6**, eabb6462 (2020)) **b** Transfer-printed Pt nanowire arrays on a transparent and flexible PET substrate. (*Sci. Adv.*, **7**, eabe9083 (2021))

References

1. Kim JM, *et al.* Conformation-modulated three-dimensional electrocatalysts for high-performance fuel cell electrodes. *Sci. Adv.* **7**, eabe9083 (2021).
2. Park TW, *et al.* Thermally assisted nanotransfer printing with sub-20-nm resolution and 8-inch wafer scalability. *Sci. Adv.* **6**, eabb6462 (2020).

[Modifications in the manuscript]

(Result, page 13) “In addition, the actual electrode size achievable with the nanotransfer printing technique that we used depends on the size of the master template wafer, and scalability up to 8 inches was already demonstrated in our previous study⁵⁷. Overall, it may be promising in gigawatt-scale H₂ production since the necessary Ir-specific power for such scale was predicted to be 50 – 100 kW g⁻¹ depending on several conditions⁵⁸, which was unprecedented until the present work.

[Comment 3] Page 6, line 131: Raman is used here with the “model” EER-incorporated SnO_x for a direct evidence of the O₂⁻, but later in the manuscript XPS is chosen. Could the author comments about the challenge of Raman characterization with nanotransferred sample? Or what is the reason?

[Response 3] We appreciate the reviewer’s insightful comment regarding the Raman characterization for providing evidence of the O₂⁻. As pointed out in the comments, we employed Raman spectroscopy technique, especially surface-enhanced Raman spectroscopy (SERS), for detecting O₂⁻ incorporated on tin oxide. SERS is an effective method that can detect molecules with a very low concentration by amplifying the signal of that specific molecule using the hot spot of the plasmonic nanostructures¹. In the fundamental characterization of the samples, the existence of O₂⁻ was confirmed by forming EER-incorporated tin oxide nanowire arrays on plasmonic nanostructures to amplify the low intrinsic signal of O₂⁻. However, since most of the plasmonic nanostructures are composed of Ag or Au, which can affect the overall electrochemical performance², in the latter part of the electrochemical evaluations, XPS was used to observe the EER rather than SERS.

References

1. Kim YJ, Lee GR, Cho EN, Jung YS. Fabrication and Applications of 3D Nanoarchitectures for Advanced Electrocatalysts and Sensors. *Adv. Mater.* **32**, 1907500 (2020).
2. Shi YF, Lyu ZH, Zhao M, Chen RH, Nguyen QN, Xia YN. Noble-Metal Nanocrystals with Controlled Shapes for Catalytic and Electrocatalytic Applications. *Chem. Rev.* **121**, 649-735 (2021).

[Comment 4] *Page 6, Line 133: “a proper initiation point”. This expression is not clear.*

[Response 4] We thank the reviewer for catching this ambiguity which is now corrected in the manuscript.

[Modifications in the manuscript]

(Result, page 6) “Moreover, the atomic ratio of O and Sn in deposited SnO_x can be controlled by selecting an initiation point of e-beam deposition, as shown in **Fig. 1d**, enabling the fabrication of SnO_x with varied EER content.”

[Comment 5] *Page 6, line 138 and Supp. Figure 5: In the caption of Supp Figure 5, it is not explained which of the species is the O₂⁻.*

[Response 5] We appreciate the reviewer’s valuable comment. We indicated the superoxides (O₂⁻) species in O 1s spectra as loosely bound oxygen with green color. However, since this indication is unclear, we modified the legend in the **Supplementary Fig. 5**.

[Modifications in the manuscript]

(Supplementary Information)

Supplementary Fig. 5. X-ray photoelectron spectroscopy (XPS) spectra of Bulk and EER contained ATO nanowire arrays with varying density of EER. The binding energies of Sn 3d spectra (a, c, e, g) and O 1s spectra (b, d, f, h) of each sample, respectively.

[Comment 6] Page 8, line 207: “highest mass activity value...(Supp. Table 2)”. The sentence is correct, but it is noticeable how the Ir loading is much lower than the other compared catalysts, and Ir mass is in the denominator of mass activity. I think the authors should also compare specific activity in the table, for a more fair comparisons.

[Response 6] We appreciate the reviewer for this suggestion. We include the specific activity, which is normalization of mass activity by electrochemically active surface area, in the **Supplementary Table 2**.

[Modifications in the manuscript]

(Result, page 8) “As shown in **Fig. 3g** and **Supplementary Table 2**, Ir/D-ATO also presents superior specific activity compared to the other catalysts and Ir-based catalysts using metal oxide supports reported previously.”

(Supplementary Information)

Supplementary Table 2. Comparison of experimental conditions (mass loading, electrolyte), mass activity, specific activity, and stability of Ir/D-ATO with previously reported Ir-based electrocatalysts using support materials in half-cell measurement.

Catalyst	Mass loading ($\mu\text{g}_{\text{Ir}} \text{cm}^{-2}$)	Electrolyte	Mass activity ($\text{A mg}_{\text{Ir}}^{-1}$)	Specific activity (mA cm^{-2})	Stability	Ref
Ir/D-ATO	2.023	0.05M H_2SO_4	5.975 (at 1.55 V)	4.542 (at 1.55 V)	Chronopo. 10 mA cm^{-2} 18 h	This work
IrNiO _x /meso-ATO	10.2	0.05M H_2SO_4	0.08 (at 1.51 V)	0.030 (at 1.51 V)	Chronopo. 1 mA cm^{-2} 20 h	1
Ir-ND/ATO	10.2	0.05M H_2SO_4	0.0698 (at 1.51 V)	0.029 (at 1.51 V)	Chronopo. 1 mA cm^{-2} 15 h	2

IrO _x /ATO	10.2	0.05M H ₂ SO ₄	0.021 (at 1.51 V)	0.008 (at 1.51 V)	Chronopo. 10 mA cm ⁻² 1.5 h	3
MW-Ir/ATO	20	1M H ₂ SO ₄	1.86 (at 1.58 V)	1.207 (at 1.58 V)	Chronopo. 10 mA cm ⁻² 15 h	4
Ir/ATO	204	0.5M H ₂ SO ₄	0.845 (at 1.48 V)	0.571 (at 1.48 V)	N/A	5
Ir _{NP} -ITO	102	0.1M HClO ₄	0.035 (at 1.51 V)	N/A	Chronopo. 10 mA cm ⁻² 2 h	6
IrO ₂ -TiO ₂ - 245	100	0.1M HClO ₄	0.070 (at 1.525 V)	0.028 (at 1.525 V)	R.Chronoam. 500 cycles (~90%)	7
IrNi NPNWs	25	0.1M HClO ₄	0.732 (at 1.53 V)	0.438 (at 1.53 V)	Chronopo. 5 mA cm ⁻² 200 min	8

[**Comment 7**] Page 9, line 225: “by holding the potential to obtain constant current densities”, maybe a typo here...

[**Response 7**] We thank the reviewer for catching this typo which is now corrected in the revised manuscript.

[Modifications in the manuscript]

(**Result, page 9**) “We also assessed the long-term stability of the Ir/D-ATO by measuring the potential to obtain constant current density together with the reference samples.”

[Comment 8] Page 9, line 233: “with such a high current density” 10 mA cm⁻² is not high, and the current chosen for the stability test of 1 mA cm⁻² shown in Figure 3b is also very low. Could the authors justify their choices?

[Response 8] We agree with the concern raised by the reviewer for the conditions of chronopotentiometry measurement to evaluate the stability of catalysts in half-cell test. We chose 1 and 10 mA cm⁻², which are the half-cell stability test conditions that are conventionally used in previous studies¹. In the half-cell measurement, the performance of the catalyst itself is estimated without any other stacked components, so even 10 mA cm⁻² is considered harsh enough^{2,3}. As already discussed in **Reviewer #1 – [Response 2]** and **Reviewer #2 – [Response 6]**, we performed additional stability test for PEMWE single-cell under high current density conditions of 0.5 and 1 A cm⁻², which are within the current state-of-the-art evaluation conditions. Under these realistic high current conditions, our Ir/D-ATO exhibited superior electrochemical durability than other candidates. Thus, we believe it is significant that the excellent performance in the half-cell test was equally effective in the PEMWE single-cell systems.

References

1. Chen FY, Wu ZY, Adler Z, Wang HT. Stability challenges of electrocatalytic oxygen evolution reaction: From mechanistic understanding to reactor design. *Joule* **5**, 1704-1731 (2021).
2. McCrory CCL, Jung S, Ferrer IM, Chatman SM, Peters JC, Jaramillo TF. Benchmarking Hydrogen Evolving Reaction and Oxygen Evolving Reaction Electrocatalysts for Solar Water Splitting Devices. *J. Am. Chem. Soc.* **137**, 4347-4357 (2015).
3. McCrory CCL, Jung SH, Peters JC, Jaramillo TF. Benchmarking Heterogeneous Electrocatalysts for the Oxygen Evolution Reaction. *J. Am. Chem. Soc.* **135**, 16977-16987 (2013).

[Comment 9] Page 9, line 251: “The tafel slope of Ir/D-ATO (49.05 mV dec⁻¹, Fig 3e) is significantly lower compared to Ir/M-ATO (51.40 mV dec⁻¹)...”. I would argue that the difference of 2 mV dec⁻¹ is not a big difference. If the authors want to use the term “significant” they should show uncertainties from reproductions. Otherwise I would suggest to simply rephrase the sentence. With S-ATO is ok, there the difference is 10 mV dec⁻¹.

[Response 9] We agree with reviewer’s comment and accept the reviewer’s suggestion. We have simply rephrased the sentence in the revised manuscript.

[Modifications in the manuscript]

(Result, page 10) “Also, the Tafel slope of Ir/D-ATO (49.05 mV dec⁻¹, **Fig. 3e**) showed a lower value compared to Ir/M-ATO (51.40 mV dec⁻¹) and Ir/S-ATO (59.33 mV dec⁻¹), indicating ~”

[Comment 10] Figure 1d: In the caption it would be useful to specify how the three regions are identified, which I guess it is from the fitting of the oxygen XPS peak. For example, otherwise it is unclear why the limit between medium and sparse is at ~150 etch seconds. Maybe the Si substrate region can also be indicated, but this last is a minor change.

[Response 10] We concur with the reviewer's comment that the specific criteria for dividing the three regions in Fig. 1d should be articulated in the caption. As the reviewer correctly pointed out, this standard is derived from the atomic ratio of O to Sn as depicted in Fig. 1d. First, for the fabrication of dense EER samples, we selected and utilized only the region where the atomic ratio of O is higher than Sn (represented by the red region in Fig. 1d). Furthermore, to fabricate sparse EER samples, we chose a plateau region exhibiting negligible variance in the atomic ratio of O to Sn as the e-beam deposition progressed (shown as the green region in Fig. 1d). Finally, we could fabricate moderate EER samples using the intermediate region (indicated by the yellow region in Fig. 1d). For all three samples, the total deposition time remained constant, revealing that the regions are divided depending on the atomic ratio of O and Sn. To accommodate the reviewer’s comment, we have added a sentence in the caption of Fig. 1d.

[Modifications in the manuscript]

(Fig. 1. Caption, page 19) “**Fig. 1 Design and prediction ~. d** XPS depth profile of an e-beam deposited SnO_x thin film on Si substrate. The three regions are divided based on the atomic percentage of O and Sn: red (where the atomic percentage of O is higher than Sn), yellow (where the atomic percentage of O intersects with Sn), and green (where the atomic percentage values of O and Sn are comparable). **e** Raman spectroscopy spectra ~”

[Comment 11] *Figure 2b): This figure panel is not clear to me. For example, the supporting figure S9a is much clearer. In my opinion, the issue with Figure 2b is the lack of labels and the fact that the single steps are not described by a short underneath text. Also the 3D structure with the alternately stacked layers is not fully shown, in contrast to figure s9a.*

[Response 11] We are grateful to the reviewer for the valuable comments. Especially, **Fig. 2** focuses on the fabricating process of the building block constituting the OER catalysts by forming Ir on the EER-contained ATO nanowire arrays and characterization of its physiochemical properties. However, we agree with the reviewer, and thus added the illustration showing the 3D-stacked array structure in the revised manuscript (**Fig. 2**). Also, to clarify the fabrication process illustrated in Fig. 2b, brief descriptions under each step were updated.

[Modifications in the manuscript]

Fig. 2 Fabrication and characterization ~. b Fabrication process of Ir/D-ATO using solvent-assisted nanotransfer printing (S-nTP) with sequential e-beam deposition and high temperature annealing. ~

(Supplementary Information)

Supplementary Fig. 10. Scanning electron microscopy (SEM) image of the final morphology of Ir/D-ATO with 500 nm scale bar.

[Comment 12] *Figure 2c and caption: In the caption the scale bar for 2c is described in the text referred to d), after point d).*

[Response 12] We thank the reviewer for catching this error which is now corrected in the caption of **Fig. 2c** of the revised manuscript.

[Modifications in the manuscript]

(Fig. 2. Caption, page 20) “Fig. 2 Fabrication and characterization ~. c TEM image with 50 nm scale bar (the inset images: SEM image with 500 nm scale bar (upper left) and SAED pattern (lower right) of Ir-/D-ATO) and d HRTEM image with 5 nm scale bar of the fabricated Ir/D-ATO. e XRD spectra ~”

[Other Modifications in the manuscript]

(Acknowledgements) The acknowledgements have been updated.

- Initial version of manuscript:

This work was supported by Korea Institute of Energy Technology Evaluation and Planning (KETEP) grant funded by the Korean government (MOTIE) (No. 20214000000650, Energy Innovation Research Center for Fuel Cell Technology). This work was supported by ~~Korea Institute of Energy Technology Evaluation and Planning (KETEP) grant funded by the Korean government (MOTIE) (No. 20213030030260)~~ and the program of Future Hydrogen Original Technology Development through the National Research Foundation (NRF) of Korea funded by the Ministry of Science and ICT (NRF-2021M3I3A1082879). This work was also supported by the National Research Foundation (NRF) of Korea funded by the Ministry of Science and ICT (NRF-2022M3H4A7046278).

- Current version of manuscript:

This work was supported by Korea Institute of Energy Technology Evaluation and Planning (KETEP) grant funded by the Korean government (MOTIE) (No. 20214000000650, Energy Innovation Research Center for Fuel Cell Technology). This work was supported by the program of Future Hydrogen Original Technology Development through the National Research Foundation (NRF) of Korea funded by the Ministry of Science and ICT (NRF-2021M3I3A1082879). This work was also supported by the National Research Foundation (NRF) of Korea funded by the Ministry of Science and ICT (NRF-2022M3H4A7046278).

REVIEWERS' COMMENTS

Reviewer #1 (Remarks to the Author):

In the revised manuscript, the authors have added some experimental results (quantification of XPS, SEM and electrochemical measurements) to support the structure and performance of Ir/D-ATO according to the reviewers' suggestions. The quality of the manuscript has been improved. However, the degradation rates at current densities of 1.0 and 0.5 A cm⁻² surprisingly appear to be smaller than that at 0.1 A cm⁻² in PEMWE. The authors should address the issue before the acceptance for publication.

Reviewer #2 (Remarks to the Author):

The authors have made extensive efforts to tackle all the scientific questions raised by myself and the other reviewers. I believe that the manuscript has considerably improved its cohesiveness and quality and consequently I can recommend its publication in its current state.

Reviewer #3 (Remarks to the Author):

The authors addressed all my comments and suggestions and modified the text accordingly, including further clarification of the EER, comments on scalability, addition of specific activity comparison, and stability at higher current densities. I recommend publication in this Journal.

■ **Point-by-point responses to the reviewer's comments.**

Reviewer #1

[General remarks] *In the revised manuscript, the authors have added some experimental results (quantification of XPS, SEM and electrochemical measurements) to support the structure and performance of Ir/D-ATO according to the reviewers' suggestions. The quality of the manuscript has been improved. However, the degradation rates at current densities of 1.0 and 0.5 A cm⁻² surprisingly appear to be smaller than that at 0.1 A cm⁻² in PEMWE. The authors should address the issue before the acceptance for publication.*

[General response] We greatly value the reviewer's helpful comment. The observed reduced degradation rate at higher current densities can be attributed to the increased Ir loadings, as indicated in the table below. As the reviewer already pointed out, it is evident that higher applied current densities lead to accelerated catalyst degradation (for the same samples with Ir loading of 47.5 μg_{Ir} cm⁻²) while increasing the Ir loading significantly enhances the overall stability of the catalyst (7.2 vs. 47.5 μg_{Ir} cm⁻²).

Ir loading (μg _{Ir} cm ⁻²)	Current density (A cm ⁻²)	Voltage degradation rate (mV h ⁻¹)
7.2	0.1	0.674
47.5* (newly updated)	0.5	0.184
47.5* (newly updated)	1.0	0.624

*The increased Ir loading for the stability test at higher current densities is mentioned on page 12 in the revised manuscript.

For instance, at 0.1 A cm⁻² (our previous characterization condition), even an extremely small Ir loading was sufficient to withstand the relatively low current density. However, to ensure stable operation at higher current densities (1 A cm⁻², requested by the reviewer) for an extended period, adopting higher Ir loadings was critical. This is because for a higher Ir loading, more surface area is available to share the applied current density, resulting in a reduced workload per unit catalytic surface.

Similar cases of applying high Ir loadings for a stability test at high current densities can be easily found in recent works, as shown in the table below. Notably, **our catalyst demonstrated excellent stability with considerably lower Ir loading** and showed promise for performing well even at higher current densities exceeding 1.0 A cm⁻². These loading conditions were described in the manuscript.

	Ir catalyst loading (mg _{Ir} cm ⁻²)	Stability on MEA	Operating temperature (°C)	Reference
Ir/D-ATO	0.0475	500 h @ 0.5 A cm ⁻² 250 h @ 1.0 A cm ⁻²	80	This work
DNP-IrNi	0.67	100 h @ 2 A cm ⁻²	90	Energy Environ. Sci. 2022 , 15, 3449–3461
Ir/Nb ₂ O _{5-x}	1.8	2000 h @ 2 A cm ⁻²	80	Angew. Chem. Int. Ed. 2022 , 61, e202212341
Ta _{0.1} Tm _{0.1} Ir _{0.8} O _{2-δ}	0.16 ^a	500 h @ 1.5 A cm ⁻²	50	Nat. Nanotechnol. 2021 , 16, 1371–1377
Sr-Ru-Ir	0.46 ^b	150 h @ 0.5 A cm ⁻²	80	J. Am. Chem. Soc. 2021 , 143, 6482–6490

^aCalculated based on the empirical formula (total catalyst loading = 0.2 mg cm⁻² (Ir ratio = 0.8))

^bCalculated based on the ICP-OES data (total catalyst loading = 2 mg cm⁻² (23 wt% Ir))

Reviewer #2

[General remarks] *The authors have made extensive efforts to tackle all the scientific questions raised by myself and the other reviewers. I believe that the manuscript has considerably improved its cohesiveness and quality and consequently I can recommend its publication in its current state.*

[General response] We appreciate the reviewer's positive evaluation of the revised manuscript. As a result of the reviewer's insightful comments and suggestions, our manuscript has been improved. We hope the final version of the manuscript successfully delivers the concept of our work to the readers of the journal.

Reviewer #3

[General remarks] *The authors addressed all my comments and suggestions and modified the text accordingly, including further clarification of the EER, comments on scalability, addition of specific activity comparison, and stability at higher current densities. I recommend publication in this Journal.*

[General response] We are grateful for the reviewer's positive assessment of the revised manuscript. The reviewer's valuable comments and suggestions enabled us to greatly improve our manuscript. Hopefully, the final version of the manuscript conveys the concept of our work effectively to the readers of the journal.